# USP17L promotes the 2-cell-like program through deubiquitination of H2AK119ub1 and ZSCAN4

Panpan Shi[1,2,10], Xukun Lu[3,4,5,6,7,10], Kairang Jin[1,2], Linlin Liu [1,2], Guoxing Yin [1,2], Wenying Wang[6,7], Jiao Yang[1,2], Lijuan Wang[6,7], Lijun Dong[6,7], Wei Xie [6,7] ✉ & Lin Liu [1,2,8,9] ✉

In mouse, minor zygotic genome activation (ZGA) precedes and is essential for major ZGA in two-cell (2C) embryos. A subset of ZGA genes (known as "2C" genes) are also activated in a rare population of embryonic stem cells (ESCs) (2C-like cells). However, the functions of the 2C genes are not fully understood. Here, we find that one family of the 2C genes, *Usp17l*, plays critical roles in transcriptional and post-translational regulation of the 2C-like state in mESCs. Specifically, USP17LE, a member of the USP17L family, deubiquitinates H2AK119ub1 and promotes the expression of *Dux* and the downstream 2C genes and retrotransposons. Moreover, USP17LE deubiquitinates and stabilizes ZSCAN4. In mouse pre-implantation embryos, *Dux* is marked by strong H2AK119ub1 except for the 1-cell and early 2-cell stages. *Usp17le* overexpression reduces H2AK119ub1 and promotes *Dux* and 2C gene activation. Thus, our findings identify USP17L as a potential regulator of the 2C program.

Zygotic genome activation (ZGA) is essential for the maternal-to-zygotic transition and mammalian embryonic development. Prior to major ZGA, when thousands of genes are activated, there is a minor wave of transcription known as minor ZGA[1,2]. In mouse, minor ZGA occurs from the S phase of the zygote to the 2-cell embryo[3], which has been shown to be essential for proper major ZGA occurring at the late 2-cell stage[3,4]. Dozens of genes are activated during minor ZGA, but their functions are not completely understood. Among them, ZSCAN4 has been shown to enhance the expression of *MERVL* and a subset of 2-cell specific genes in mouse embryonic stem cells (ESCs)[5], and promote telomere elongation and genome stability[6–9]. DUX in mice and DUX4 in humans can activate a subset of minor ZGA genes in ESCs[10–12], although loss of *Dux* alone is compatible with mouse ZGA and

embryonic development[13–15]. Recently, OBOX family genes, which include both maternal factors and proteins expressed during minor ZGA, have been reported to be essential for both mouse major and minor ZGA as well as mouse development beyond the 2-cell stage[16].

2C-like cells (2CLCs) represent a rare subpopulation (1%–5%) of ESCs that express 2C-specific markers, such as *Zscan4* and endogenous retrovirus (*MERVL*)[6,17,18]. 2CLCs and early 2-cell embryos share unique characteristics of transcriptomes, chromatin accessibility landscapes, increased global histone mobility, and regulatory networks[1,10,17,19–22]. In addition, 2CLCs also recapitulate 2-cell embryos in their capacity to contribute to extra-embryonic tissues[17]. Hence, 2CLCs represent an excellent model for understanding ZGA and totipotency[22,23].

[1]State Key Laboratory of Medicinal Chemical Biology, Nankai University, Tianjin, China. [2]Frontiers Science Center for Cell Responses, College of Life Sciences, Nankai University, Tianjin, China. [3]State Key Laboratory of Reproductive Medicine and Offspring Health, Center for Reproductive Medicine, Institute of Women, Children and Reproductive Health, Shandong University, Jinan, Shandong, China. [4]National Research Center for Assisted Reproductive Technology and Reproductive Genetics, Shandong University, Jinan, Shandong, China. [5]Key Laboratory of Reproductive Endocrinology (Shandong University), Ministry of Education, Jinan, Shandong, China. [6]Tsinghua-Peking Center for Life Sciences, Beijing, China. [7]Center for Stem Cell Biology and Regenerative Medicine, MOE Key Laboratory of Bioinformatics, New Cornerstone Science Laboratory, School of Life Sciences, Tsinghua University, Beijing, China. [8]Tianjin Union Medical Center, Nankai University, Tianjin, China. [9]Haihe Laboratory of Cell Ecosystem, Tianjin, China. [10]These authors contributed equally: Panpan Shi, Xukun Lu. ✉e-mail: xiewei121@tsinghua.edu.cn; liulin@nankai.edu.cn

The DUB/USP17 family proteins were initially identified in mice as deubiquitination enzymes that are involved in cell growth and viability, DNA damage response, and embryogenesis[24–26]. *Usp17l*, which includes *Usp17la-e*, is a subfamily of *Dub/Usp17* residing in a tandemly repeated sequence on chromosome 7 in mice[24,27]. Notably, *Usp17l* is highly expressed in minor ZGA[28] and in 2CLCs[29–32]. Its overexpression can improve the efficiency of SCNT[33]. However, the exact function of *Usp17l* and its working mechanisms remain unknown. In this study, we show that *Usp17l* plays critical roles in the 2-cell-like state through deubiquitination of H2AK119ub1 to facilitate the expression of *Dux*, and through deubiquitination and stabilization of ZSCAN4.

## Results

### USP17L promotes 2C-gene activation in ESCs

*Usp17l* gene family includes *Usp17la (Dub1)*, *Usp17lb (Dub1a)*, *Usp17lc (Dub2)*, *Usp17ld (Dub2a)*, and *Usp17le (Dub3)* that share stretches of identical nucleotide sequences in the coding regions (Fig. 1A). As reported[28,33], *Usp17l* genes were activated during minor ZGA (early 2-cell) and the transcripts peaked during major ZGA (late 2-cell) followed by a rapid decline afterward (Fig. 1B, left). Ribo-seq data[34] showed they were also actively translated during this period (Fig. 1B, right). As *Usp17l* was also highly expressed in 2CLCs[29], we utilized *Zscan4*-positive (+) ESCs as a model to study the potential function of *Usp17l*[21,23,35,36]. We first sorted *Zscan4+* and *Zscan4−* cells by flow cytometry of tdTomato-ZSCAN4 ESCs (Fig. 1C and Supplementary Fig. 1A). *Usp17l* was expressed at higher levels in *Zscan4+* ESCs than that in *Zscan4−* cells (Fig. 1D), as reported[29]. Elevated USP17L protein levels in *Zscan4+* ESCs were also confirmed by western blot (Supplementary Fig. 1B) using an antibody that recognizes all family members ("Methods").

To explore the potential functions of *Usp17l*, we first attempted to knock them down in ESCs. Based on the common region of the five *Usp17l* subfamily members (Fig. 1A), two shRNAs were designed and transfected into ESCs, from which two stable knockdown cell lines (KD4, KD9) (Fig. 1E, F) were selected for subsequent experiments. We confirmed the substantial downregulation of *Usp17l* at both the RNA and protein levels (Fig. 1E, F). The *Usp17l* knockdown ESCs displayed typical ESC morphology (Fig. 1G). Interestingly, RNA-seq analysis revealed that pluripotency genes such as *Nanog*, *Klf2*, *Tbx3*, *Dppa3*, and *Esrrb* were upregulated following *Usp17l* knockdown (Fig. 1H, left). *Pou5f1* (also known as *Oct4*) expression was substantially upregulated in one of the clones (KD4, Fig. 1H, left). Western blot showed the upregulation of both OCT4 and NANOG in both cell lines (Supplementary Fig. 2A). By contrast, *Usp17l* knockdown substantially reduced the expression of *Dux*, *Zscan4*, and other 2C genes (Fig. 1H, left, and Supplementary Fig. 2B), and the proportion of *Zscan4+* cells (Fig. 1I). Transcription of *MERVLs* and other retrotransposons was also decreased (Supplementary Fig. 2C). Hence, USP17L promotes the activation of 2C genes in ESCs. The upregulation of pluripotency genes upon *Usp17l* KD was similar to that in *Dux* KO[37] or *Zscan4* KD mESCs[5]. Acquisition of totipotency or expanded developmental potential is often accompanied by downregulation of pluripotency genes for reasons that are not very clear[38,39]. How USP17L suppresses pluripotency genes in mESCs warrants further investigation.

We then examined whether overexpression of *Usp17l* can promote the transcription of 2C genes. Intriguingly, overexpression of *Usp17le* substantially increased the expression of *Zscan4* and the proportion of *Zscan4+* cells revealed by flow cytometry and immunofluorescence microscopy (Fig. 1J and Supplementary Fig. 2D, E). *Zscan4* was only slightly upregulated when other *Usp17l* members were overexpressed (Supplementary Fig. 2D). The differential activation capacity appeared to correlate with their cellular localization, as immunofluorescent showed that USP17LE was clearly localized in the nucleus, unlike other USP17L members which were present mainly in the cytoplasm and peri-nucleus regions (Supplementary Fig. 2F). Due to the lack of antibodies that can distinguish each family member, we

used an anti-Flag antibody to probe Flag-tagged USP17L proteins in ESCs. Although we could not exclude the possibility that this difference may arise from overexpression and may not reflect the properties of endogenous USP17L, we chose to use USP17LE for the majority of overexpression experiments. While pluripotency genes were downregulated, expression of *Dux*, *Zscan4* family, *Tcstv1/Tcstv3*, and other 2C-genes, as well as the *Usp17l* family itself, were notably elevated in *Usp17l* overexpressed ESCs (Fig. 1H, right). Furthermore, *MERVL* and *MT2* were also upregulated by *Usp17le* overexpression (Fig. 1K), suggesting that USP17L is sufficient to activate the 2C program. Of note, our data show that *Esrrb* was upregulated upon *Usp17l* depletion and downregulated upon *Usp17le* overexpression in mESCs (Fig. 1H). As ESRRB can upregulate *Usp17le* in coordination with the coactivator NcoA1 in mESCs[40,41], these data suggest possible negative regulatory feedback between *Esrrb* and *Usp17le* in mESCs.

Consistent with the notion that ZSCAN4 is involved in telomere lengthening[6], the relative telomere length estimated by qPCR[42] was decreased upon *Usp17l* knockdown (Supplementary Fig. 2G), as confirmed by telomere quantitative fluorescence in situ hybridization (Q-FISH) (Fig. 1L and Supplementary Fig. 2H)[43] and Southern blot (Supplementary Fig. 2I). Such shortening of telomere length is not due to perturbation of the expression of telomere genes, such as *Terc* and *Tert*, which were unaffected after *Usp17l* knockdown (Supplementary Fig. 2J). Along with the shortened telomeres, chromosome breakage was also detected in *Usp17l* knockdown cell lines (Supplementary Fig. 2H, arrows), suggesting increased genome instability. As immunofluorescence showed no localization of USP17L at telomeres marked by TRF1 (Supplementary Fig. 2K), these data raise a possibility that USP17L may affect telomeres through ZSCAN4. In summary, USP17LE promotes the transcription of 2C genes in ESCs.

### USP17LE activates 2C gene expression through *Dux* by deubiquitinating H2AK119ub1 at its locus

We then explored how USP17L regulates 2C genes. As *Dux* is a major inducer of the 2C state[6,17,44–47], we tested whether *Dux* is involved in the elevated transcription of 2C genes following *Usp17le* overexpression. Indeed, in *Usp17le* overexpressed ESCs, *Dux* knockdown partially reduced the transcription of 2C genes, including *Usp17l*, *Zscan4*, *Tcstv1*, and *MERVL* (Fig. 2A, B), raising a possibility that *Usp17le* may promote the transcription of the 2C genes through both DUX dependent and independent pathways. We then asked if the expression of *Dux* may be regulated by USP17L. *Dux* was downregulated upon *Usp17l* knockdown (Fig. 2C) and was upregulated in *Usp17le* overexpressed cells (Fig. 2D), indicating that *Dux* is regulated by USP17L at the transcriptional level. Given *Usp17l* is known to be a target of DUX[10]. The reciprocal regulation between DUX and USP17L may represent a positive feedback loop that facilitates the rapid induction of 2C genes.

We then asked how USP17L regulates *Dux*. It has been reported that USP17 family members act as deubiquitinating enzymes, and USP17LE has the core structural site of the deubiquitination enzymes[27] (Supplementary Fig. 3A, red). In particular, USP proteins (e.g., USP16 and USP21) have been shown to be involved in the regulation of H2AK119ub1[48,49], a repressive mark deposited by PRC1[50]. Interestingly, H2AK119ub1, as well as the closely related repressive mark H3K27me3[51], was highly enriched at the *Dux* locus in ESCs (Fig. 2E), which further increased in *Usp17l* KD cells (Fig. 2E, right). It is worth noting that as the *Dux* locus exists as tandem repeats[52], multi-mapping reads were included when studying chromatin marks in this region. In *Usp17l* KD ESCs, H2AK119ub1 and H3K27me3 were globally increased (Fig. 2E and Supplementary Fig. 3B). H2AK119ub1 increased at a large number of gene promoters ($n = 3149$) (fold change of RPKM $\geq 1.5$) in *Usp17l* KD ESCs, but only decreased at a limited number of genes ($n = 30$) (Fig. 2F, left). A similar trend, but to a lesser extent, was observed for H3K27me3 (increased, $n = 1930$; decreased, $n = 1366$) (Fig. 2F, right). Notably, the increase of

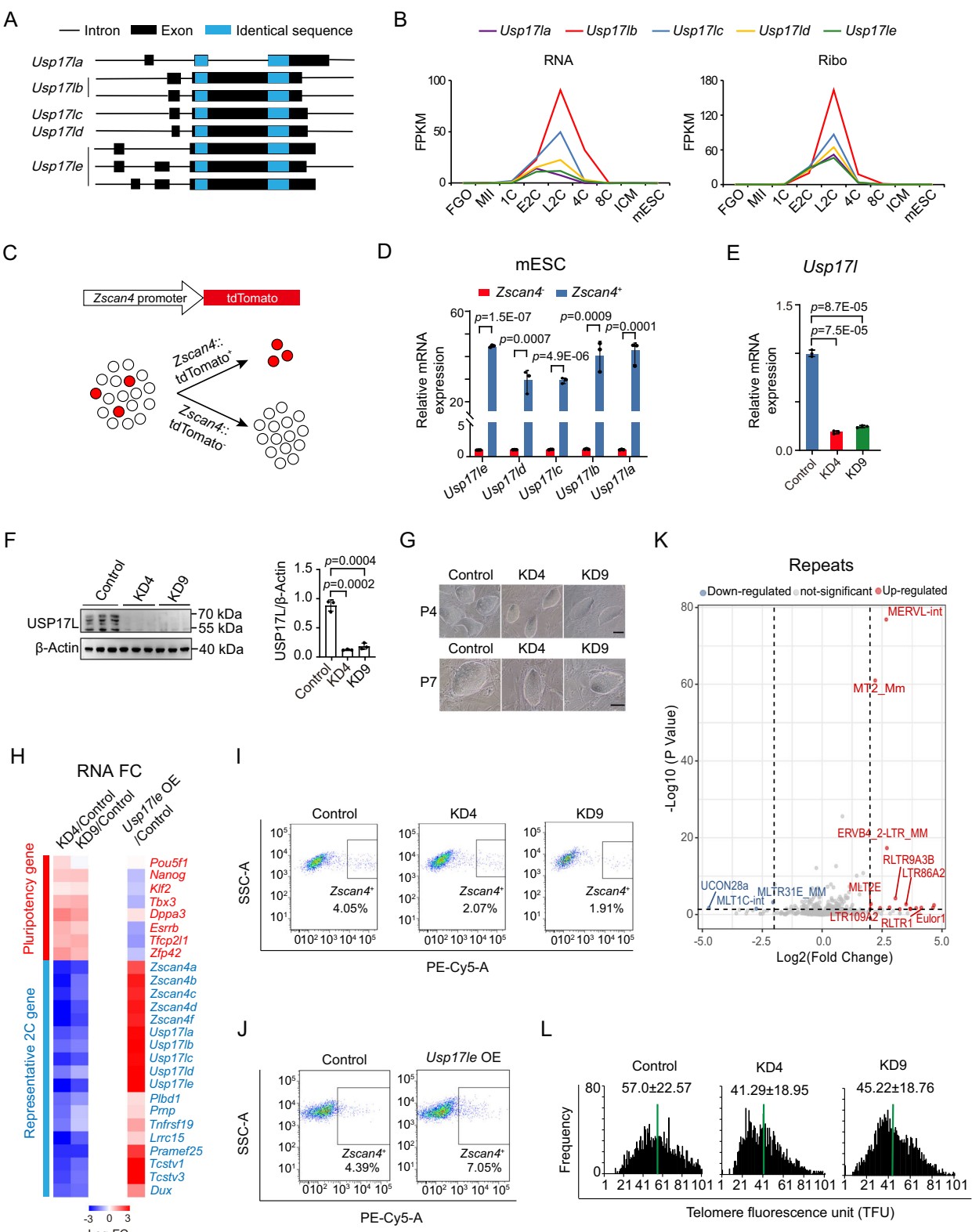

H2AK119ub1 (Fig. 2F, left) and H3K27me3 (Fig. 2F, right) at *Dux* ranked number 7 and number 1, respectively, among all genes. Given *Dux* activates *Usp17l*[10,12], the possible positive feedback may also enhance the changes of H2AK119ub1 and H3K27me3 at the *Dux* locus. Among 2C genes, only *Dux* was strongly enriched for H2AK119ub1 and H3K27me3 (Fig. 2F, red). These data indicate that USP17L may regulate *Dux* through regulating H2AK119ub1 near its locus, and the

downregulation of other 2C genes may be achieved indirectly (e.g., in part through *Dux* and other factors).

Apart from *Dux*, genes exhibiting increased promoter H2AK119ub1 and H3K27me3 were mainly involved in development and cell differentiation (Fig. 2F, bottom), consistent with their functions in regulating developmental genes[51]. Genes with decreased H2AK119ub1 were not enriched for any GO terms, likely due to the small number

**Fig. 1 | *Usp17l* regulates the 2C program in mouse ESCs. A** Schematic depicting the *Usp17l* family genes based on the nucleotide sequences. Exons and introns are represented in boxes and lines, respectively. The identical sequences shared by all family members are highlighted in blue. There are two isoforms of *Usp17b* and three isoforms of *Usp17e*. The *Usp17le* and *Usp17lb* isoforms used in this paper are both isoform 1. **B** Line charts showing the expression of *Usp17l* family genes in early embryos. RNA-seq and Ribo-seq (indicating translation) data are from published data[34,62]. **C** Schematic showing the sorting of *Zscan4* positive (+) and negative (−) cells. **D** Bar chart showing the expression of *Usp17l* family genes in *Zscan4*+ and *Zscan4*− ESCs measured by qPCR (3 biological replicates). The levels in *Zscan4*− ESCs serve as controls in each group. Data are shown as means ± SD (two-tailed Student's *t*-test). **E** Bar chart showing the expression of *Usp17l* in control and knockdown ESCs measured by RT-qPCR using a pair of primer that target all *Usp17l* genes. KD4 and KD9 represent two cell lines with effective knockdown efficiency (3 biological replicates). Data are shown as means ± SD (two-tailed Student's *t*-test). **F** Western blot showing USP17L protein levels in control and *Usp17l* knockdown cell lines detected using an antibody that target all family members. The relative levels of USP17L determined by densitometry of the protein bands are shown on the right (3 biological replicates). Data are shown as means ± SD (two-tailed Student's *t*-test). **G** The morphology of *Usp17l* knockdown ESCs on passage 4 and passage 7. Scale bar, 100 μm. **H** Heatmaps showing the changes of pluripotency genes (red) and representative 2 C genes (blue) in *Usp17l* knockdown (left) and *Usp17le* overexpression (right) ESCs. **I** Flow cytometry sorting of *Zscan4*+ cell populations in control and *Usp17l* knockdown ESCs. **J** Flow cytometry sorting of *Zscan4*+ cell populations in control and *Usp17le* overexpression ESCs. **K** Scatter plot showing the expression changes of retrotransposons following transient overexpression of *Usp17le* (2 biological replicates). **L** Histogram showing the distribution of relative telomere length shown as telomere fluorescence intensity unit (TFU) in control and *Usp17l* knockdown ESCs analyzed by TFL-TELO software. Green lines indicate median telomere length. The average length ± SD is shown on the top.

($n = 30$). Genes with decreased H3K27me3 upon *Usp17l* KD were involved in protein transport, spermatogenesis, DNA damage response, and apoptosis (Fig. 2F, bottom right). The increase of H2AK119ub1 and H3K27me3 was not associated with gene expression changes globally (Supplementary Fig. 3C, grey, $R = −0.06$ to $−0.02$), including those at developmental genes (Supplementary Fig. 3C, black), consistent with the fact that these genes are already repressed in WT ESCs. These data suggest that while H2AK119ub1 and H3K27me3 were globally altered in *Usp17l* KD ESCs, the changes at the *Dux* locus are among the most pronounced ones. Altogether, these data suggest that USP17L deubiquitinates H2AK119ub1 and promotes the expression of *Dux* and other 2C genes.

## USP17LE deubiquitinates H2AK119ub1 to regulate 2C genes

We then asked if USP17L can indeed deubiquitinate H2AK119ub1. We first confirmed that USP17L KD caused an increase of H2AK119ub1 and H3K27me3 in Western blot in ESCs (Fig. 3A), consistent with the ChIP-seq result (Fig. 2F and Supplementary Fig. 3B). Conversely, overexpression of USP17LE, but not a catalytic mutant with a mutation in a conserved site (C60A, denoted USP17LE^C60A)[27], could deubiquitinate H2AK119ub1 in mESCs, as shown by Western blot (Fig. 3B, red arrow, and Supplementary Fig. 4A). Supporting such deubiquitination activity, H2AK119ub1 levels decreased in *Usp17l*-overexpressing ESCs in immunofluorescence (Fig. 3C, D). H3K27me3 also tended to decrease in *Usp17l* overexpressed ESCs (Fig. 3D), agreeing with the notion that H2AK119ub1 can recruit PRC2[53]. Consistently, *Zscan4*+ cells exhibited lower levels of H2AK119ub1 compared with those of *Zscan4*− ESCs (Fig. 3E, F), as shown previously[21]. Interestingly, H3K9me2/3 levels were decreased upon USP17L KD in ESCs (Fig. 3A), suggesting that USP17L may directly or indirectly regulate H3K9me2/3. Taken together, these data indicate that USP17L can deubiquitinate H2AK119ub1.

We further tested whether USP17L regulates 2C gene expression through H2AK119ub1. We first treated ESCs with a specific inhibitor (PRT4165) of H2AK119ub1[54], and then analyzed the transcription levels of the 2C genes. H2AK119ub1 gradually decreased with increasing treatment time of PRT4165 (Supplementary Fig. 4B, C). Consistent with our hypothesis, transcription of 2C genes such as *Zscan4* and *Tcstv1* was increased shortly (1 h) following the inhibition of H2AK119ub1 (Supplementary Fig. 4D). ZSCAN4 protein levels also increased at a later time point (3 h) (Supplementary Fig. 4E). By contrast, OCT4 also decreased although to a lesser extent (Supplementary Fig. 4B, C). These data imply that H2AK119ub1 may repress the 2C genes, presumably through *Dux* inhibition. Importantly, inhibition of H2AK119ub1 by PRT4165 in *Usp17l* knockdown cell lines rescued the expression of 2C genes, such as *Tcstv1*, *Zscan4*, and *MERVL* (Fig. 3G, H). Taken together, these data suggest that USP17L can deubiquitinate H2AK119ub1 and further activate 2C genes including *Dux*.

## USP17L promotes the deubiquitination and stabilization of ZSCAN4

Besides H2AK119ub1, we asked if USP17L may also have non-histone substrates. Interestingly, it was reported that ZSCAN4 was subjected to ubiquitination-dependent degradation in a human cancer cell line Tu167 by an E3 ubiquitin ligase RNF20[55]. Therefore, we sought to examine if USP17L regulates ZSCAN4 protein. Treatment of ESCs with MG132, a proteasome inhibitor, led to rapid accumulation of ZSCAN4 protein (Fig. 4A), suggesting that ZSCAN4 is under active degradation in ESCs. Encouragingly, overexpression of USP17LE increased ZSCAN4 protein levels and the ratio of *Zscan4*+ cell population in ESCs (Fig. 4B, C). Treatment with the protein synthesis inhibitor cycloheximide (CHX) reduced ZSCAN4 protein levels in ESCs, which can be mitigated by USP17LE overexpression (Fig. 4D). To ask if USP17L directly deubiquitinates ZSCAN4, we overexpressed these two proteins in ESCs and found that USP17LE interacted with ZSCAN4 and reduced ubiquitination of ZSCAN4 (Fig. 4E). The global ubiquitination level appeared to decrease upon USP17LE overexpression (Fig. 4E, the 2nd lane). We cannot rule out the possibility that USP17LE also regulates other histone or non-histone targets. Importantly, overexpression of USP17LE, but not the catalytic mutant USP17LE^C60A, increased ZSCAN4 protein levels (Fig. 4F), suggesting that USP17LE stabilizes ZSCAN4 in part through deubiquitinating ZSCAN4.

To further define the key regions of USP17LE in the recognition of ZSCAN4, various truncated USP17LE mutants were overexpressed in ESCs (Fig. 4G and Supplementary Fig. 5A). Specifically, USP17LE^Δ408−506, but not others, including USP17LE^Δ458−506, abrogated the protective effect of USP17LE on ZSCAN4 (Compare Fig. 4H, D and Supplementary Fig. 5A), suggesting an essential protection function of amino acids 408−457, which was embedded in a hypervariable (HV) domain (365−461aa) that may confer substrate specificity[56]. Analysis using FoldUnfold[57] revealed that this sequence consisted of a disordered region (408−448 aa, Fig. 4I, shaded in cyan), and a relatively stable region (449−457 aa) (Fig. 4I, shaded in pink). The stable region (EVELDLPVD) contained two aspartate residues that potentially account for substrate recognition, as inferred based on the substrate interaction site of USP7[58−60]. Indeed, mutation of these two sites (denoted USP17LE^D453A/D457A) affected the binding of USP17LE with ZSCAN4 (Supplementary Fig. 5B) and abrogated the protection of ZSCAN4 proteins (Fig. 4J, K). Consistently, overexpression of USP17LE^D453A/D457A can no longer increase the ratio of *Zscan4*+ cells (Supplementary Fig. 5C). These results suggest that ZSCAN4 protein can be degraded by ubiquitin-mediated proteasome degradation and that USP17LE protects ZSCAN4 by deubiquitination.

## Overexpression of USP17L promotes removal of H2AK119ub1, *Dux* expression and 2C program in mouse embryos

We then asked if *Usp17l* and H2AK119ub1 may regulate *Dux* and the 2C program in mouse embryos. Reanalysis of published data[61] showed

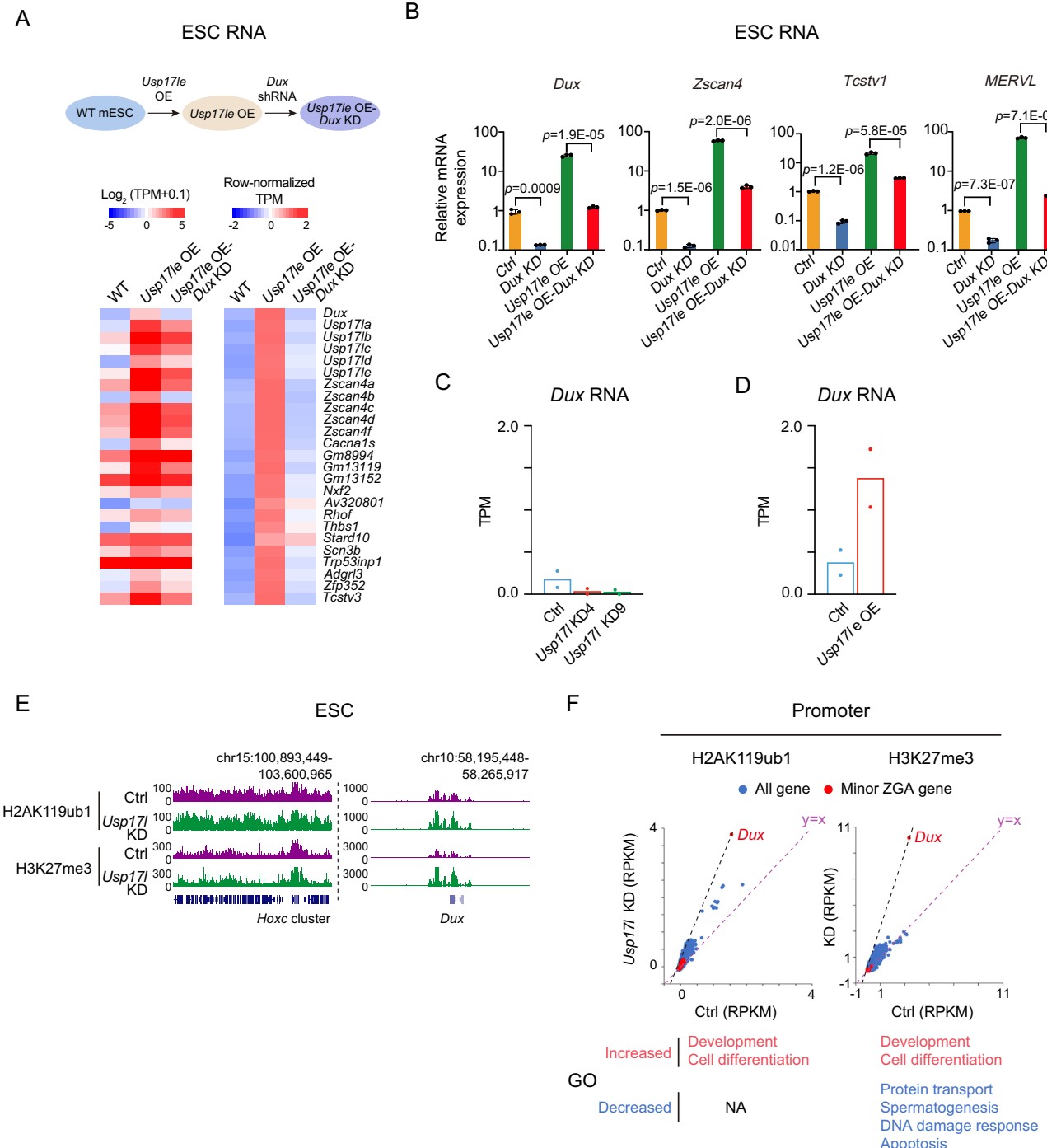

**Fig. 2 | USP17LE decreases H2AK119ub1 to derepress *Dux*. A** Top, schematic showing the establishment of *Usp17le* OE and *Usp17le* OE-*Dux* KD mESCs. *Usp17le* OE-*Dux* KD mESCs are established by knocking down *Dux* in *Usp17le* OE mESCs. Bottom, heatmap showing the changes of 2C genes in WT, *Usp17le* OE, and *Usp17le* OE-*Dux* KD mESCs. Both the raw (left) and row-normalized TPM (right) are shown. **B** RT-qPCR result showing the expression of representative 2C-genes (*Dux*, *Zscan4*, *Tcstv1*, and *MERVL*) in control, *Dux* knockdown, *Usp17le* OE, and *Usp17le* OE-*Dux* knockdown ESCs (3 biological replicates). Data are shown as means ± SD (two-tailed Student's *t*-test). **C** Bar chart showing the transcription level (TPM) of *Dux* after *Usp17l* knockdown in ESCs. Individual values are shown for each replicate (2 biological replicates). **D** Bar chart showing the transcription level (TPM) of *Dux* after *Usp17le* overexpression in ESCs. Individual values are shown for each replicate (2 biological replicates). **E** The UCSC genome browser view showing the global H2AK119ub1 and H3K27me3, and that at the *Dux* locus (multi-mapping allowed) in WT and *Usp17l* KD mESCs. **F** Scatter plots showing H2AK119ub1 (left) and H3K27me3 (right) at gene promoters in WT and USP17L KD mESCs. The *Dux* locus is shown. Note its value is high as *Dux* is a muti-copy gene, and muti-mapping reads were used. The purple dashed line indicates the RPKM values when they show equal values between control and *Usp17l* KD mESCs. The black dashed line indicates H2AK119ub1/H3K27me3 changes at gene promoters when they are equal to that at the *Dux* locus upon USP17L KD. The Gene Ontology terms of genes exhibiting increased/decreased H2AK119ub1 and H3K27me3 are shown at the bottom.

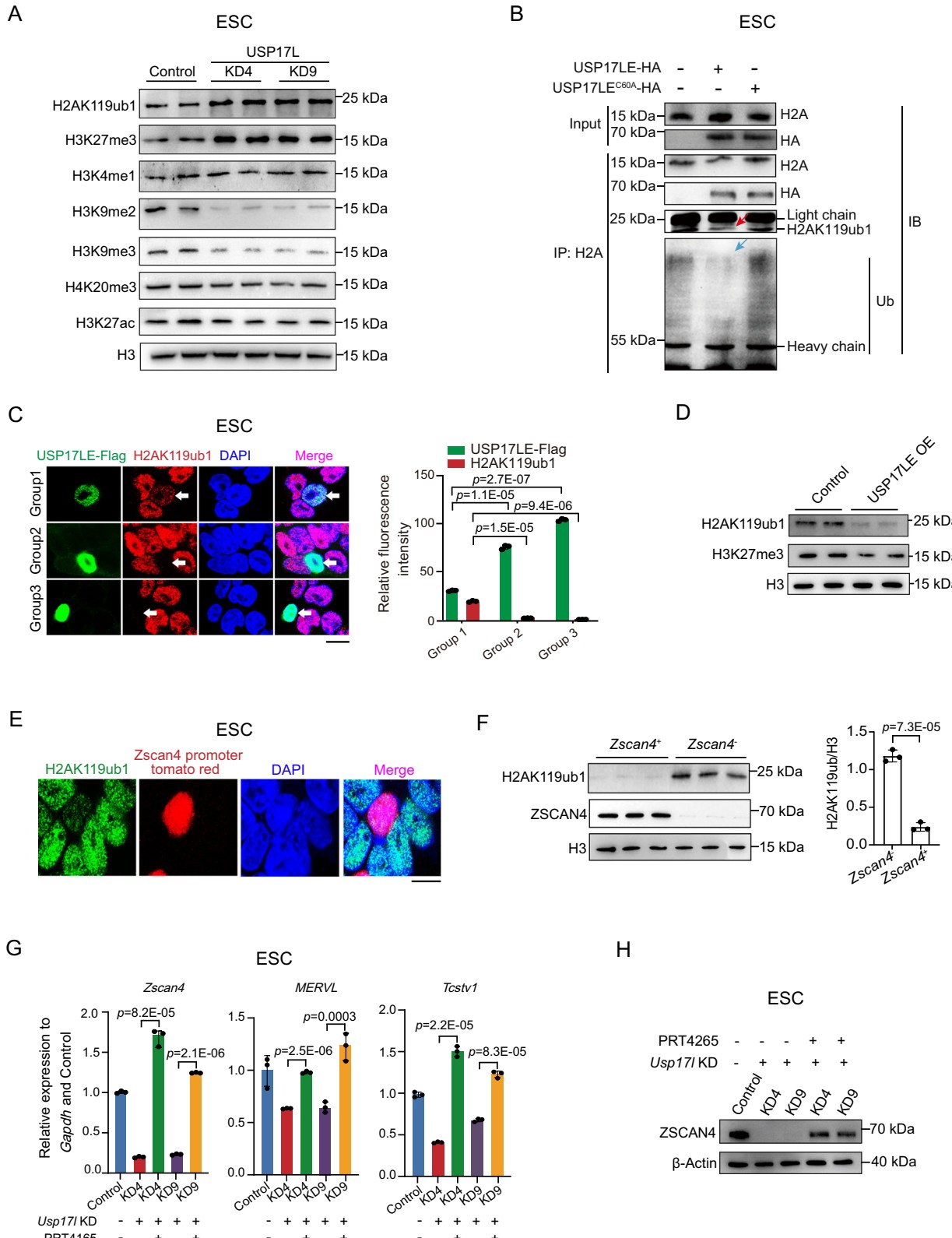

that H2AK119ub1 was highly enriched near *Dux* in mouse oocytes, decreased at the 1-cell and early 2-cell stages when *Dux* was activated (Fig. 5A, B)[62], and then increased and remained at high levels at the late 2-cell stage until the blastocyst stage (Fig. 5A). Therefore, the dynamics of H2AK119ub1 at the *Dux* locus negatively correlated with the expression changes of *Dux* during mouse pre-implantation development. H3K27me3 was also enriched in mouse oocytes, decreased after

fertilization, but was not restored until in ICM (Fig. 5C). The slower restoration of H3K27me3 was consistent with the lack of de novo H3K27me3 at promoters in mouse pre-implantation embryos[63]. These data also indicate that H3K27me3 may be not a major repressor of *Dux* in mouse pre-implantation embryos.

We then asked if USP17L can also reduce H2AK119ub1 and promote the expression of *Dux* and 2C program in mouse embryos. We

**Fig. 3 | USP17LE promotes 2C gene transcription by deubiquitinating H2AK119ub1. A** Western blot showing the levels of various modified histones in control and *Usp17l* knockdown ESCs (3 biological replicates). **B** Western blot showing the levels of ubiquitin and H2AK119ub1 in the immunoprecipitates pulled down by anti-H2A antibody in control, USP17LE- and USP17LE[C60A]- overexpressing ESCs. The heavy chain and light chain of IgG are indicated. Note the decrease of ubiquitin (blue arrow) and H2AK119ub1 (red arrow) when USP17LE is overexpressed (2 biological replicates). **C** Immunofluorescence staining of H2AK119ub1 (red, stained with anti-H2AK119ub1 antibody) and USP17LE (green, stained with anti-Flag antibody) in three groups of ESCs. Scale bar, 10 μm. The three groups refer to USP17LE with different overexpression levels (green). The relative fluorescence intensity of H2AK119ub1 and USP17LE in the same cell in each group is shown on the right (3 biological replicates). Data are shown as means ± SD (two-tailed Student's *t*-test). **D** Western blot showing the levels of H2AK119ub1 and H3K27me3 after USP17LE overexpression (3 biological replicates). **E** Immunofluorescence of H2AK119ub1 (green, stained with anti-H2AK119ub1 antibody) and ZSCAN4 (red, tdTomato). Scale bar, 10 μm. **F** Western blot showing the protein levels of H2AK119ub1 in *Zscan4*+ and *Zscan4*- ESCs. The relative levels of H2AK119ub1 determined by densitometry of the protein bands were shown on the right (3 biological replicates). Data are shown as means ± SD (two-tailed Student's *t*-test). **G** RT-qPCR result of the relative expression of *Zscan4*, *MERVL*, and *Tcstv1* in control, *Usp17le* knockdown ESCs with or without H2AK119ub1 inhibitor PRT4165. PRT4165 was added at 50 μM and treated for 3 h (3 biological replicates). Data are shown as means ± SD (two-tailed Student's *t*-test). **H** Western blot showing the expression of ZSCAN4 in control, *Usp17le* knockdown ESCs with or without H2AK119ub1 inhibitor PRT4165. PRT4165 was added at 50 μM and treated for 3 h (3 biological replicates).

injected *Usp17l* siRNAs that targeted all *Usp17l* gene family members in mouse zygotes at the pronuclear stage 3 (PN3) (Fig. 6A). We confirmed the efficient *Usp17l* KD in mouse late 2C embryos (Fig. 6B, left). Spike-in normalized Stacc-seq[4] for H2AK119ub1 and CUT&RUN[64] for H3K27me3, however, showed that unlike that in ESCs, *Usp17l* KD did not affect H2AK119ub1 or H3K27me3 globally at the late 2C stage (Fig. 6C, left, "KD", and 6D, the 1st and 3rd columns), as also confirmed by immunostaining (Fig. 6E). At the *Dux* locus, H2AK119ub1 was largely unaffected upon *Usp17l* KD (Fig. 6C, right, "KD" for "H2AK119ub1", and 6D, the 1st column, red). H3K27me3 remained overall low with moderate increase upon *Usp17l* KD (Fig. 6C, right, "KD" for "H3K27me3", and 6D, the 3rd column). *Usp17l* KD did not have an apparent impact on mouse pre-implantation development (Fig. 7A, B, compare "KD" with "Ctrl") or major ZGA (Fig. 7C, left, and 7D). Minor ZGA showed only moderate downregulation in *Usp17l* KD late 2C embryos (Fig. 7C, left, and 7D), as exemplified by *Zscan4* family genes (Fig. 7E). *Dux* was evidently downregulated (Fig. 7E). Given that H2AK119ub1 was not affected near *Dux* upon *Usp17l* KD, we speculate that the expression of *Dux* may be also regulated by additional mechanisms, at the transcriptional or post-transcriptional levels (such as RNA stability)[44,65–67].

To overcome the possible redundancy of USP17L and other deubiquitinases such as USP16, a maternal protein that can deubiquitinate H2AK119ub1 in oocytes and is present in both oocytes and early embryos[68], we overexpressed *Usp17le* in mouse embryos by injecting *Usp17le* mRNA in mouse PN3 zygotes. Considering the presence of *Usp16* and endogenous *Usp17l*, *Usp17le* was expressed at high levels (Fig. 6B, right). *Usp17le* OE caused a substantial global reduction of H2AK119ub1 and H3K27me3 (Fig. 6C, left, "OE" for "H2AK119ub1" and "H3K27me3", and 6D, the 2nd and 4th columns), which was confirmed by immunostaining (Fig. 6E). At the *Dux* locus, while H3K27me3 remained low upon *Usp17le* OE (Fig. 6C, right, "OE" for "H3K27me3", and 6D, the 4th column, red), H2AK119ub1 decreased markedly (Fig. 6C, right, "OE" for "H2AK119ub1", and 6D, the 2nd column, red). *Usp17le* OE embryos developed normally to the 2C stage, but showed partial delay at the 4C stage, and complete arrest at the 8C-morula stage (Fig. 7A, B, compare "OE" with "Ctrl"). Despite the largely normal maternal RNA decay at the late 2C stage (Fig. 7C, right), minor ZGA genes, including other *Usp17l* members, *Zscan4* family genes and *Dux*, were all upregulated (Fig. 7C, right, and 7D, E). Additionally, several classes of repeats, including LINE1 (L1Md_F, L1Md_T) and MERVL (MT2_Mm, MERVL_int) that are normally activated in minor ZGA were also upregulated upon *Usp17le* OE (Fig. 7F). Major ZGA genes were partially decreased possibly due to slight developmental delay at the late 2C stage (Fig. 7C, right, and 7D). It should be noted that the upregulation of minor ZGA genes was unlikely due to developmental delay, as the activation were dramatically increased at levels much higher than those in control embryos at the late 2C stage (Fig. 7D) when the levels of minor ZGA peaked in WT embryos (Supplementary Fig. 6A), as also exemplified by the

*Zscan4* gene family (Supplementary Fig. 6B). Therefore, consistent with the observations in ESCs, these results indicate that USP17L-mediated removal of H2AK119ub1 can promote *Dux* expression and 2C program in mouse embryos.

## Discussion

2C genes have attracted great interests in the field of stem cell, development and gene regulation. These genes are closely linked to totipotency in 2-cell embryos and 2CLCs, for reasons that still remain largely unknown. They are also subjected to unique regulation as these genes undergo a rapid induction during the 1-cell and 2-cell stages, before declining soon after the 2-cell stage. Despite being a prominent marker gene for 2-cell embryos and 2CLCs, the function of USP17L remains poorly understood. Moreover, H2AK119ub1 was reported to decrease in 2CLCs compared to ESCs[21], yet the underlying mechanisms remain elusive. Here, we report a role of *Usp17l* in the activation of the 2C program in mESC and possibly in embryos as well. Our data support a model that USP17LE deubiquitinates H2AK119ub1 near *Dux*, thus derepressing *Dux* and downstream 2C genes, including *Zscan4* and retrotransposons such as *MERVL* (Fig. 7G). Given that DUX can also activate *Usp17l* genes[10,12], this may represent a positive feedback loop (Fig. 7G). Such positive feedback may facilitate the rapid induction of 2C genes in a short time window, which then need to be swiftly shut down at the late 2C stage when major ZGA occurs through negative regulators such as DUXBL[69]. Persist expression of *Dux* has been shown to arrest embryos at the 2-cell stage[13,66]. In addition, ZSCAN4 can also be protected and stabilized through deubiquitination by USP17L (Fig. 7G). These data suggest that USP17L can regulate the 2C program both at the transcriptional and post-translational levels. Of note, we do not exclude the possibility that USP17L may have other targets besides H2AK119ub1 and ZSCAN4.

In contrast to that in mESCs, *Usp17l* KD in mouse embryos has only a moderate effect on H2AK119ub1, minor ZGA, and pre-implantation development. Several scenarios could be envisioned to explain the difference. First, USP17L may be redundant with other deubiquitinases such as USP16. Second, the function of *Dux* is also possibly compensated by other regulators of minor ZGA, given that *Dux* KO only had a mild effect on ZGA in mouse embryos[13–15]. For example, OBOX has been shown to regulate many targets of DUX[16,70] and OBOX4 redundantly drives ZGA in the absence of *Dux*[71]. Of note, *Usp16* and *Obox* genes were absent or only lowly expressed in mESCs compared to those in mouse oocytes or pre-implantation embryos[16,68], which may account for the lack of such redundancy in mESCs. However, we cannot exclude the possibility that USP17L functions differently in these two contexts. The conclusion that USP17L regulates the 2C program both at the transcriptional and post-translational levels is primarily based on data in mESCs. The roles of USP17L in mouse early embryogenesis warrant further investigation, which may require combined disruption of USP17L and other potentially redundant factors. We also expect that the targets of USP17L may not be limited to histones and ZSCAN4. In

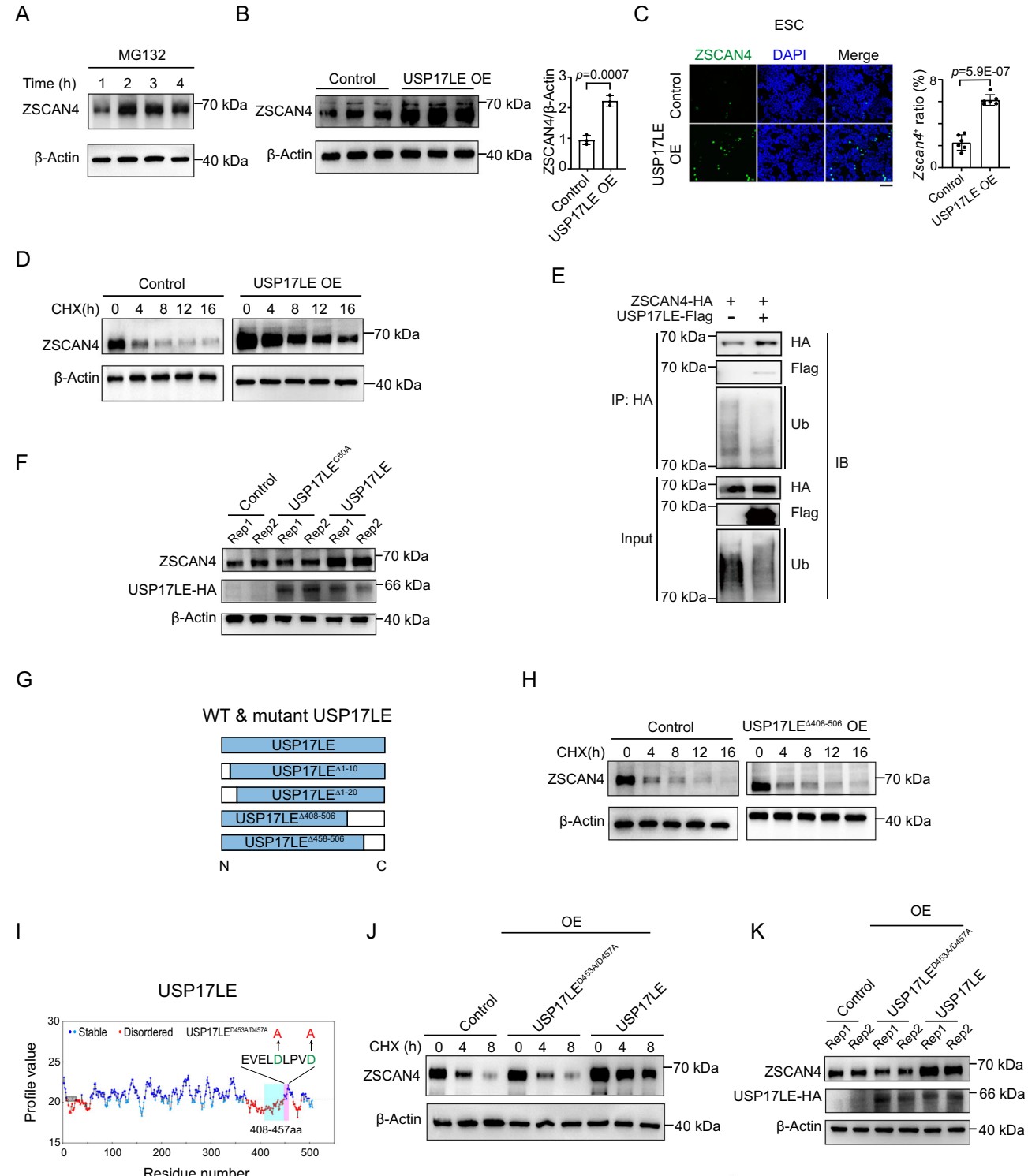

sum, our findings identify USP17L as a key regulator of the 2C genes in mESCs, paving the way for future studies of the regulatory circuitry underlying the 2C program, ZGA, and totipotency.

## Methods

### Mice

C57BL/6J mice were purchased from Beijing Vital River. PWK/PhJ mice were purchased from Jackson Laboratory. Mice were maintained under specific pathogen-free conditions with a 12 h-12 h light-dark cycle in a 20–22 °C and humidity 55 ± 10% environment. All animals were taken care and experiments were performed according to the guidelines of

the Institutional Animal Care and Use Committee (IACUC) of Tsinghua University, Beijing, China (protocol number 17-XW1).

### ES cell lines and cell culture

Two mouse ES cell lines, J1 and E3, were used in this study. The E3 ES cell line was derived from C57BL/6 × 129S6 blastocysts and tested for pluripotency using standard chimera and tetraploid embryo complementation assays. E3 ESCs were cultured on MEF feeder cells, while J1 ESCs were cultured on gelatin without feeders. The culture medium of ESCs consisted of knockout DMEM (Gibco, 10829018) and 15% FBS (HyClone, SH30070.03E) supplemented with penicillin (100 U/ml) and

**Fig. 4 | USP17LE stabilizes ZSCAN4 via deubiquitination. A** Western blot showing the accumulation of ZSCAN4 after treatment with MG132 (10 μM) in ESCs. Samples were collected every 1 h (3 biological replicates). **B** Left, western blot showing the levels of ZSCAN4 protein after *Usp17le* overexpression for 48 h. Right, the relative level of ZSCAN4 determined by densitometry of the protein bands (3 biological replicates). Data are shown as means ± SD (two-tailed Student's *t*-test). **C** Left, immunofluorescence staining for ZSCAN4 in control and USP17LE overexpressing ESCs. DNA is stained with DAPI. Right, statistics for the ratio of *Zscan4⁺* cells in control and USP17LE overexpressing ESCs. Scale bar, 100 μm (3 biological replicates). Data are shown as means ± SD (two-tailed Student's *t*-test). **D** Western blot showing ZSCAN4 protein levels in USP17LE overexpressing ESCs compared with control after treatment with cycloheximide (CHX, 50 μM) for indicated time (3 biological replicates). **E** Western blot showing the ubiquitination level of ZSCAN4 following USP17LE overexpression in ESCs. IP was performed using anti-HA antibody. Immunoblot was performed using anti-HA, anti-Flag, and anti-ubiquitin (Ub) antibodies (2 biological replicates). **F** Western blot showing the levels of ZSCAN4

protein in control, USP17LE^C60A or USP17LE overexpressing ESCs. β-Actin is used as loading control. **G** Schematic showing various truncated mutants of USP17LE. The blank boxes denote the truncated regions (3 biological replicates). **H** Western blot showing ZSCAN4 protein levels in control and USP17LE^Δ408–506 overexpressing ESCs after treatment with cycloheximide (CHX, 50 μM) for indicated time (3 biological replicates). **I** The stable and disordered region of USP17LE analyzed by FoldUnfold software. The disordered region (408–448 aa) in amino acids 408–457 is shaded in cyan, and the stable region (449–457 aa, EVELDLPVD) is shaded in pink. The stable region contains two aspartate (D) residues that potentially account for substrate recognition were mutated to alanine (A). **J** Western blot showing the levels of ZSCAN4 protein in control, USP17LE^D453/457 and USP17LE overexpressing ESCs after treatment with cycloheximide (CHX, 50 μM) for indicated time. β-Actin is used as loading control (3 biological replicates). **K** Western blot showing the levels of ZSCAN4 protein in control, USP17LE^D453/457 and USP17LE overexpressing ESCs. β-Actin is used as loading control (3 biological replicates).

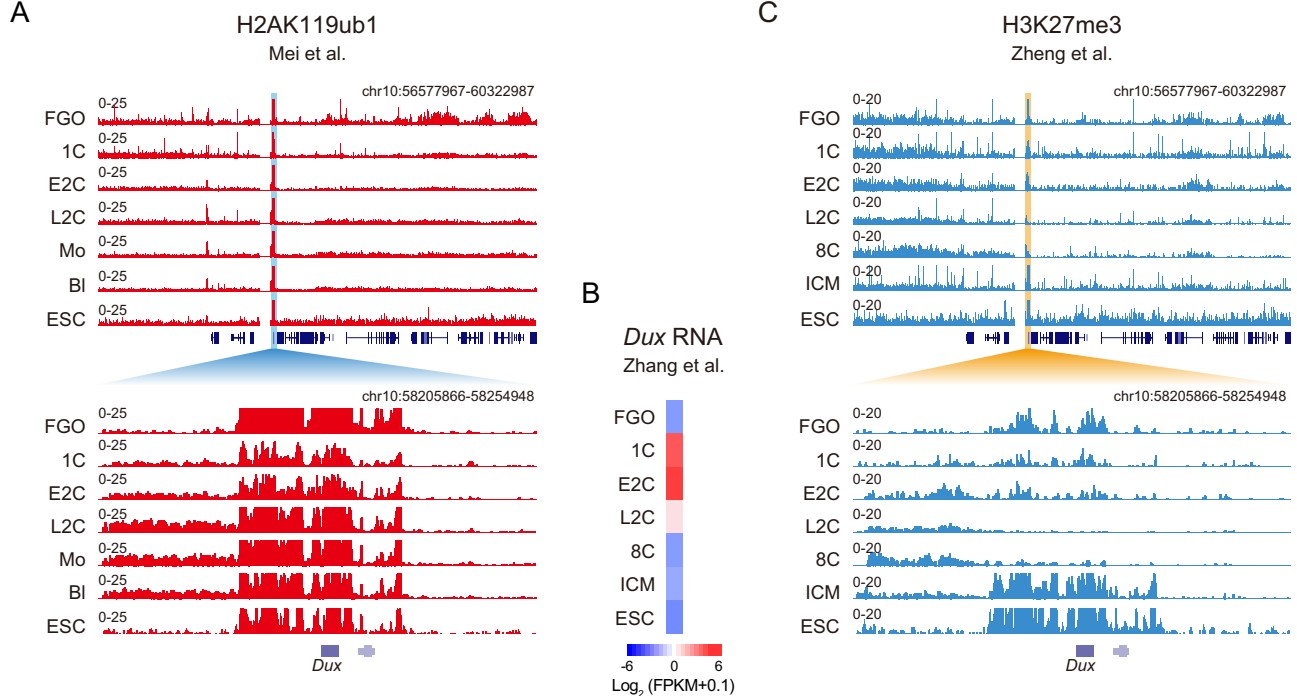

**Fig. 5 | H2AK119ub1 is enriched at the *Dux* locus in mouse embryos. A** The UCSC genome browser view showing global H2AK119ub1 (top) and that near *Dux* locus (bottom) in mouse early embryos. H2AK119ub1 data are from published data[61]. FGO full-grown oocytes; 1C, 1-cell; E2C, early 2-cell, L2C, late 2-cell; Mo morula, Bl blastocyst. H2AK119ub1 in mESC is shown as a comparison. **B** Heatmap showing the expression of *Dux* during mouse pre-implantation development. RNA-seq data are

previously published[62]. **C** The UCSC genome browser view showing global H3K27me3 (top) and that near *Dux* locus (bottom) in mouse early embryos. H3K27me3 data are from published data[63]. FGO, full-grown oocytes; 1C, 1-cell; E2C, early 2-cell, L2C, late 2-cell; 8C, 8-cell; ICM, inner cell mass. H3K27me3 in mESC is shown as a comparison.

streptomycin (100 μg/ml) (Gibco,15140122), LIF (1000 U/ml) (Millipore, ESG1107), nonessential amino acids, β-mercaptoethanol (0.1 mM) (Sigma, M3148-250ML), and L-glutamine (1 mM) (Gibco, 25030081). ESCs were cultured at 37 °C in a humidified 5% CO₂ incubator. The culture medium was changed every day and passaged every two days by replacement with fresh medium. The E3 ES cell line was used for *Usp17l* KD and OE experiments. The J1 ES cell line, which carries the *Zscan4*-tdTomato reporter, was used for mutation experiments, flow analysis, and ubiquitination experiments.

**Generation of Usp17l knockdown (KD) ESCs by shRNA**
Two shRNAs (ShRNA1-F: GATCCGAGTTTCTCATGTTCACCTTTCAAGA-GAAGGTGAACATGAGAAACTCTTTTTTG; ShRNA1-R: AATTCAAAAAA-GAGTTTCTCATGTTCACCTTCTCTTGAAAGGTGAACATGAGAAACTCG; ShRNA2-F: GATCCGTGGAGGTCTCAGATCAAGTTCAAGAGACTTGATCT

GAGACCTCCACTTTTTTG; ShRNA2-R: AATTCAAAAAAGTGGAGGTCT-CAGATCAAGTCTCTTGAACTTGATCTGAGACCTCCACG) were designed for the *Usp17l* knockdown experiment using online design tool (https://rnaidesigner.thermofisher.com/rnaiexpress/sort). The single strands were annealed (95 °C 30 s, 72 °C 2 min, 37 °C 2 min, 25 °C 2 min) into double-stranded shRNAs, which were then inserted into the knockdown plasmid RNAi-Ready pSIREN. The concentration and purity of the transfected plasmids were measured to ensure the transfection efficiency. The plasmids were transfected into ESCs by the Lonza nuclear transfer apparatus. 24 h later, the medium was replaced with that containing 1.5 μg/ml puromycin (Lifetechnologies, A1113803) for screening for 24 h. Single clones were then picked 48 h later for culture and identification. Two cell lines with effective knockdown were selected for subsequent experiments, which were further treated with 1.5 μg/ml puromycin for one week before use. ShRNA for *Dux* knockdown (Dux-

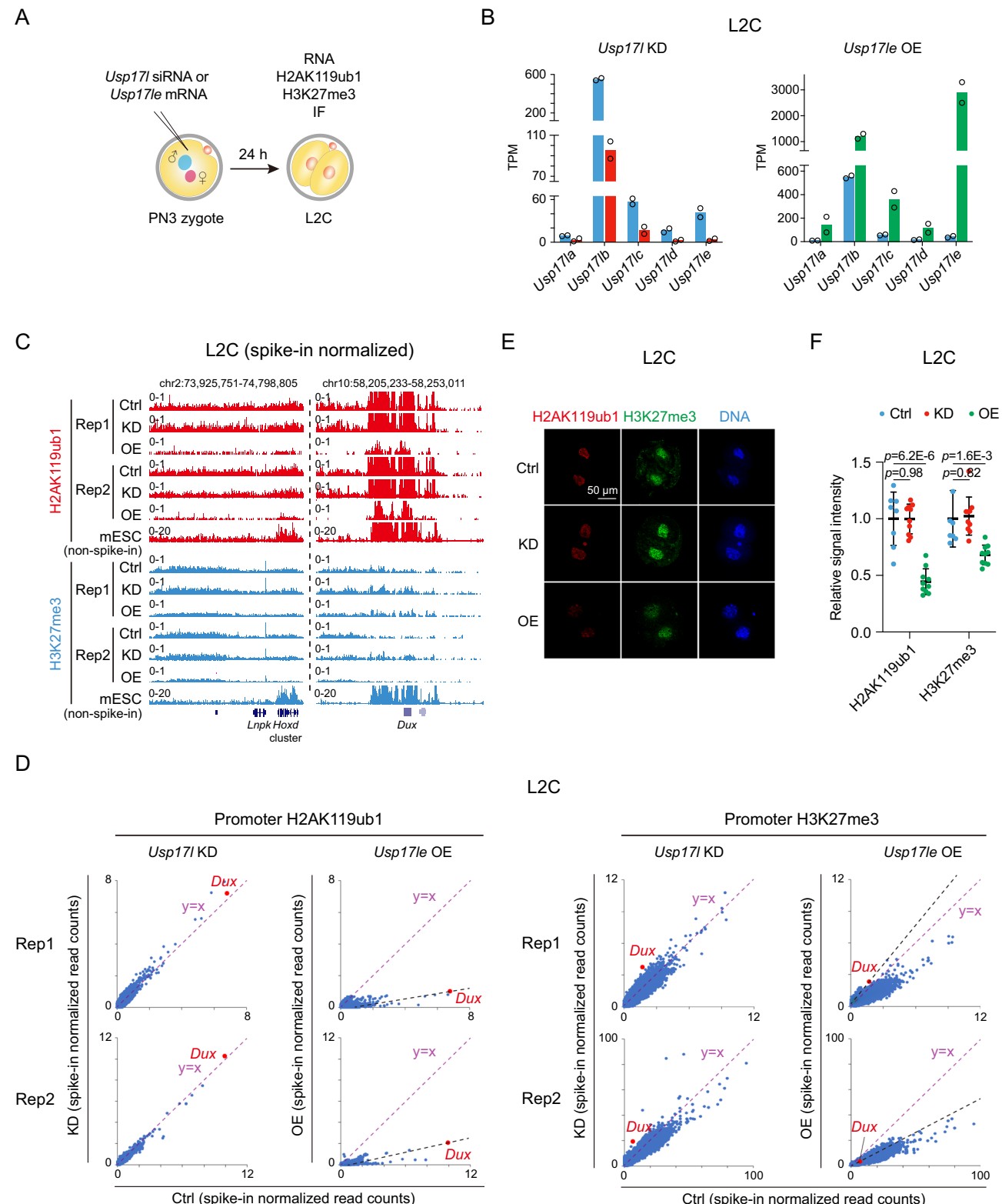

ShRNA1-F: GATCCGAGGTTCCCAGGACAGCTTACTTTCAAGAGAAGTAA
GCTGTCCTGGGAACCTTTTTTTG; Dux-ShRNA1-R: AATTCAAAAAAAGG
TTCCCAGGACAGCTTACTTCTCTTGAAAGTAAGCTGTCCTGGGAACCT
CG) was constructed as reported.

### Primers and reagents
All primers were designed using Primer 5 software. Quantitative reverse transcription PCR (RT-qPCR) primers (Supplementary Table 1) were used to detect the expression changes of 2-cell genes. Primers for overexpression (Supplementary Table 2) and for USP17LE mutation (Supplementary Table 3) were used to obtain the targets, which were cloned into pcDNA3.1-3 × FLAG vector. The *Usp17le* and *Usp17lb* isoforms used in the overexpression experiment were both isoform 1 as shown in Fig. 1A. Primer STAR Max DNA Polymerase (TAKARA, R045A) was used for PCR. The resultant vectors were transfected into ESCs with Hieff Trans® Liposomal Transfection Reagent (Yeasen,

**Fig. 6 | USP17L reduces H2AK119ub1 at the *Dux* locus in mouse embryos.**
**A** Schematic showing injection of siRNA cocktail targeting all *Usp17l* family genes or *Usp17le* mRNAs into mouse PN3 zygotes and examination of RNA, H2AK119ub1, and H3K27me3 at the late 2C stage. **B** Bar charts showing the transcription levels (TPM) of *Usp17l* family genes after *Usp17l* KD (left) and *Usp17le* OE (right) in mouse late 2C embryos (2 biological replicates). **C** The UCSC genome browser visualization showing spike-in normalized H2AK119ub1 and H3K27me3 at *Hoxd* (left) and *Dux* (right) upon *Usp17l* KD (KD) and *Usp17le* OE (OE) in mouse late 2C embryos. H2AK119ub1 and H3K27me3 in mESCs (non-spike-in) are shown as controls.
**D** Scatter plots showing the spike-in normalized read counts of H2AK119ub1 (left) and H3K27me3 (right) at gene promoters in *Usp17l* KD and *Usp17le* OE late 2C

embryos. Data from two replicates are shown. *Dux* is highlighted in red. The purple dashed line indicates that the spike-in normalized read counts when they are equal between control and KD/OE L2C embryos. The black dashed line indicates that H2AK119ub1 or H3K27me3 changes at gene promoters when they are equal to that at the *Dux* locus. **E, F** Representative immunostaining images (**E**) and jitter plot (**F**) showing the changes of H2AK119ub1 and H3K27me3 upon *Usp17l* KD and *Usp17le* OE in mouse late 2C embryos. Scale bar, 50 μm. The signal intensity of the control was set to 1.0. Bar = mean ± SD. The numbers of late 2C examined were 8 (Ctrl), 10 (KD), and 10 (OE) for both H2AK119ub1 and H3K27me3. *p* values (two-tailed Student's *t*-test) are shown.

40802ES02) according to the manufacturer's instructions. RNA was extracted 48 h later using RNeasy Plus Mini Kit (QIAGEN, 74134).

### Mutation of USP17LE
USP17LE$^{\Delta1-10}$ has a deletion of 10 amino acids at the N-terminus; USP17LE$^{\Delta1-20}$ has a deletion of 20 amino acids at the N-terminus; USP17LE$^{\Delta407-506}$ removes 99 amino acids from the C-terminus; USP17LE$^{\Delta458-506}$ removes 49 amino acids from the C-terminus. The 408–457 amino acids were analyzed using the FoldUnfold software[57]. According to the binding information of USP7[58–60], the two aspartic acids, D453 and D457, were mutated. The sequences encoding the truncated USP17LE were amplified using specific primers (Supplementary Table 3) by PCR. In addition, the mutated bases were introduced into the primers, followed by overlap extension PCR to amplify the DNA encoding USP17LE containing the mutated sites (USP17LE$^{D453/457}$).

### Coimmunoprecipitation (Co-IP)
Collected cells were lysed with 500 μl ice-cold NETN buffer (20 mM Tris-HCl pH 8.0, 1 mM EDTA, 500 mM NaCl, 0.5% NP-40) supplemented with 1 mM PMSF and 1× protease inhibitor cocktail by gentle pipetting. The lysate was incubated at 4 °C for 30 min, followed by centrifugation at 4 °C at maximum speed for 10 min. A total of 500 μl supernatant was collected, of which 50 μl was reserved as input control. The remaining supernatant was equally divided into two aliquots in new microcentrifuge tubes. Rabbit IgG (negative control) and anti-USP17L antibody were added to each aliquot, respectively. The mixtures were incubated with rotation at 4 °C overnight. The next day, after pre-washed three times (8 min each time) with NETN buffer containing PMSF and protease inhibitors, protein A beads were resuspended in 100 μl NETN buffer and equally split into the IgG- and USP17L-treated samples. After rotation at 4 °C for 3 h, beads were collected by centrifugation (8000 × *g* at 4 °C for 1 min). Beads were washed six times with 1 mL ice-cold NETN buffer (8 min per wash, 4 °C with rotation). After the final wash, the precipitate was resuspended in 80 μl NETN buffer and boiled with 5× SDS loading buffer at 98 °C for 5 min. Proteins were resolved by SDS-PAGE and visualized using immunoblotting.

### Immunofluorescence microscopy
The samples were fixed with 4% paraformaldehyde (PFA) at 4 °C for 20 min, washed three times with PBS, permeabilized for 25 min and blocked at room temperature for 2 h. The primary antibody was incubated overnight at 4 °C, and the secondary antibody was incubated at room temperature for 1 h. The antibodies used were as follows: rabbit anti-USP17L (1:50, Produced by GenScript, targeted the short peptide HRQSEPTSEDSSPIC), rabbit anti-ZSCAN4 (1:200, Sigma-Aldrich, AB4340), rabbit anti-H2AK119ub1 (1:200, CST, 8240S), and mouse anti-H3K27me3 (1:200, CST, 9733S). The DNA was stained with DAPI (1:200, Sigma, D9542-1MG). The images were acquired by a CCD camera equipped with laser scanning confocal microscopy (Leica, TCS SP8) or epifluorescence imaging (Zeiss Axio Imager).

### Western blot
Whole-cell protein extracts were prepared with NP40 lysis buffer (Thermo Scientific, FNN0021) supplemented with the protease inhibitor PMSF and cocktail (Sigma, P8340-1ML). The prepared protein samples were then subjected to SDS-PAGE analysis, and were transferred to the PVDF membrane in a wet transfer mode in an ice bath. The PVDF membrane was blocked with 5% skim milk at room temperature for 2 h. The primary antibody was incubated at 4 °C overnight, and the secondary antibody was incubated at room temperature for 1 h. The antibodies were as follows: mouse anti-FLAG (1:2000, Sigma–Aldrich, F1804-50UG), rabbit anti-ZSCAN4 (1:2000, Sigma–Aldrich, AB4340), mouse anti-OCT4 (1:2000, Santa Cruz, sc-5279), rabbit anti-H2AK119ub1 (1:2000, CST, 8240S), rabbit anti-HA (1:2000, Yeasen, 30702ES20), rabbit anti-ACTIN (1:5000, Abclonal, AC026), rabbit anti-H3 (1:5000, Abcam, ab1791–100 μg), rabbit anti-H3K9me2 (1:2000, Abcam, ab1220), rabbit anti-H3K9me3 (1:2000, Abcam, ab8898), rabbit anti-H3K27me3 (1:2000, Millipore, 07–449), rabbit anti-H3K4me (1:2000, Abcam, ab176877), rabbit anti-H4K20me3 (1:2000, Millipore, 07-463-S), rabbit anti-Nanog (1:2000, Abcam, ab80892), goat anti-mouse HRP (1:10000, ABWAYS TECHNOLOGY W, AB0102) or goat anti-rabbit HRP (1:10000, ABWAYS TECHNOLOGY W, AB0101). Anti-USP17L polyclonal antibody against a short peptide HRQSEPTSEDSSPIC shared by USP17LA-E was produced by GenScript.

### Flow cytometry
The cells were resuspended in FACS Buffer (0.1% BSA in PBS) and filtered through a 40 μm filter (Falcon) to remove large cell clumps, followed by analysis using a FACS Aria II flow cytometer (BD, 85 μm nozzle). The area-scaling factor was set, and forward scatter (FSC)-A and side scatter (SSC)-A were used to exclude large-sized cell structures or debris, with SSC-W set to avoid contamination by doublets or triplets. Finally, the appropriate fluorescence channels corresponding to the excitation light (594 nm) were used for cell sorting.

### Telomere restriction fragment (TRF) measurement
TRF analysis was performed using a commercial kit (TeloTAGGG Telomere Length Assay, Roche, 12209136001). Cells were pretreated with RNase A and proteinase K (PCR Grade, Roche Life Science, 03115879001), followed by extraction using phenol:chloroform:isoamyl alcohol. A total of 3 μg DNA was digested overnight with MboI endonuclease (NEB, R0147S) at 37 °C and underwent electrophoresis through 1% agarose gels in 0.5 × TBE at 14 °C for 16 h at 6 V/cm with an initial pulse time of 1 s and end pulse in 12 s using a CHEF Mapper pulsed-field electrophoresis system (Bio-Rad). The gel was blotted and probed using reagents in the kit. Telomere length was quantified by Telo Tool software.

### Telomere quantitative fluorescence in situ hybridization (Q-FISH)
Telomere length and function (telomere integrity and chromosome stability) were estimated by telomere Q-FISH as described previously[43]. In brief, cells were incubated with 0.3 μg/ml nocodazole for 3 h to enrich metaphases. Metaphase-enriched cells were exposed to

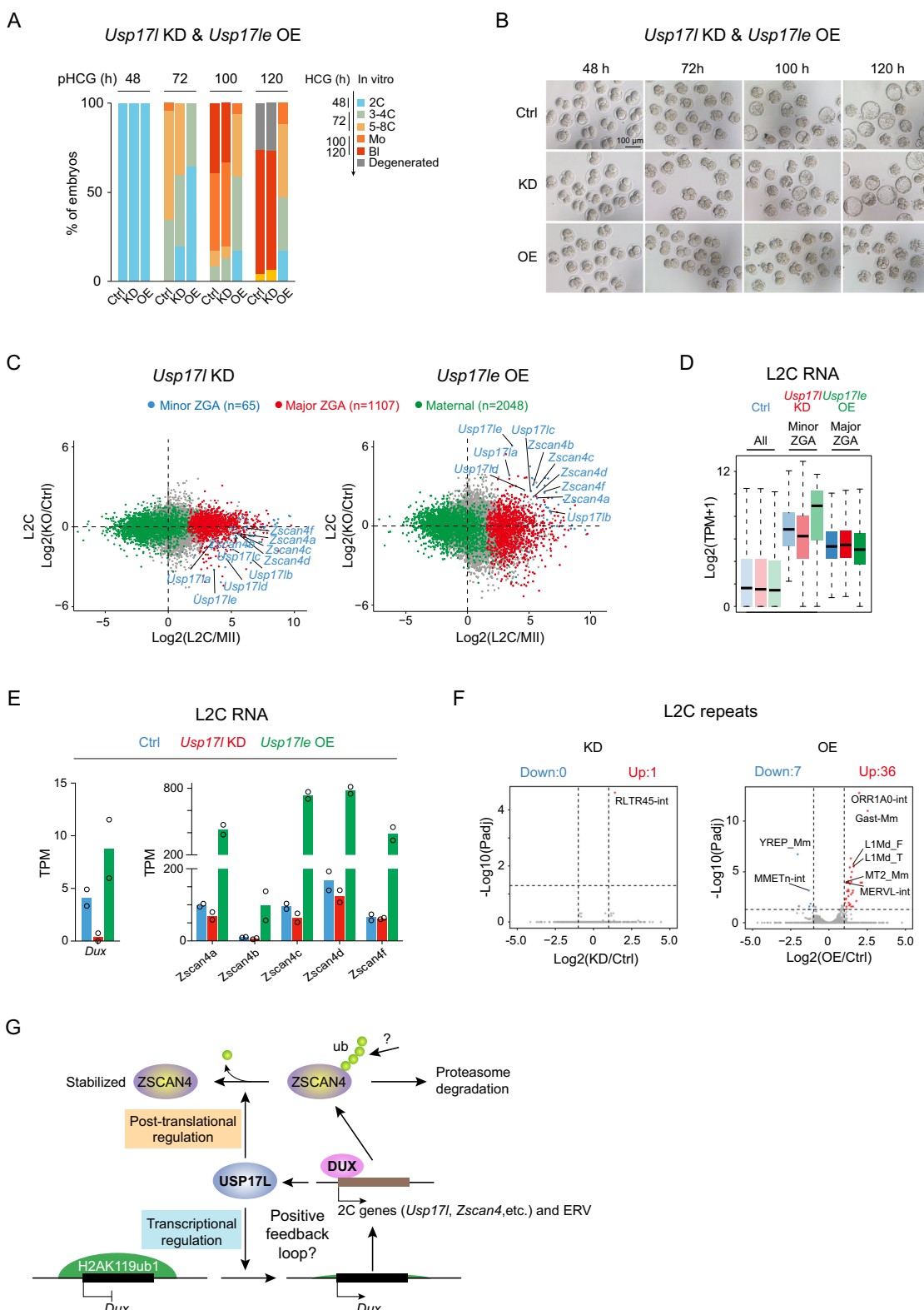

hypotonic treatment with 75 mM KCl solution, fixed with methanol: glacial acetic acid (3:1), and spread onto clean slides. Telomeres were denatured at 80 °C and hybridized with a Cy3-labelled (CCCTAA)3 peptide nucleic acid (PNA) telomere probe (0.5 μg/ml) (Panagene, Korea). Chromosomes were stained with 0.5 μg/ml DAPI. Fluorescence from chromosomes and telomeres was digitally imaged on a Zeiss Axio Imager Z2 with Cy3/DAPI filters using AxioCam and AxioVision

software. Telomere length shown as telomere fluorescence intensity was integrated using the TFL-TELO program (a gift kindly provided by Peter Lansdorp).

**siRNA and mRNA microinjection**
For *Usp17l* knockdown, four siRNA duplexes each at a concentration of 100 μM were mixed in equal volume (the final concentration for each

**Fig. 7 | USP17L promotes *Dux* expression and 2C program in mouse embryos.**
**A** Bar chart showing the developmental ratios of mouse control, *Usp17l* KD, and
*Usp17le* OE pre-implantation embryos. The expected developmental stages of
mouse embryos in vitro at the indicated time point are shown on the right. **B** The
development of mouse control, *Usp17l* KD and *Usp17le* OE pre-implantation
embryos. Scale bar, 50 μm. **C** Scatter plots showing the changes of maternal
(green), minor ZGA (blue), and major ZGA (red) genes upon *Usp17l* KD (left) or
*Usp17le* OE (right) in late 2C embryos. **D** Box plot showing the expression of minor
and major ZGA genes in *Usp17l* KD and *Usp17le* OE late 2 C embryos (2 biological
replicates). Global gene expression ("All") is shown as a control. **E** Bar charts
showing the expression (TPM) of *Dux* (left) and *Zscan4* (right) in *Usp17l* KD and

*Usp17le* OE late 2C embryos (2 biological replicates). **F** Scatter plots showing the
expression changes of repeats in *Usp17le KD* (left) and *Usp17le* OE (right) late 2C
embryos. **G** Schematic showing the working model of USP17L in regulating the 2C
program. At the transcriptional level, USP17LE decreases the level of H2AK119ub1
near the *Dux* locus, promoting the expression of *Dux* and the downstream 2C
genes, such as *Usp17l*, *Zscan4*, and *MERVL*. Reducing H2AK119ub1 at the *Dux*
locus and activating *Dux* expression by USP17L likely represents a positive feedback loop
between DUX and USP17L, as DUX can also activate *Usp17l* genes[10,12]. At the post-
translational level, USP17L stabilizes ZSCAN4 by decreasing the ubiquitination
levels of ZSCAN4, attenuating its degradation by proteasome.

siRNA was 25 μM) to target all *Usp17l* family genes and were micro-
injected into mouse zygotes at the pronucleus stage 3 (PN3).

(#1, sense: AAACUCAUGGGCAUCUUCCUG, antisense: GGAAGAU
GCCCAUGAGUUUCU;

#2, sense: UCCAAAUAUGUCAUGAAUGGG, antisense: CAUUC
AUGACAUAUUUGGAGG;

#3, sense: AGUAUCAUCCAUCUUGUACCA, antisense: GUACAAG
AUGGAUGAUACUAA;

#4, sense: ACAUAGAAGAGCACAUAGGCA, antisense: CCUAU-
GUGCUCUUCUAUGUGC;). For *Usp17le* overexpression, T7 promoter
was first added to the *Usp17le* coding sequence (see Supplementary
Table 3 for PCR primers). *Usp17le* PCR product was then purified and
used for in vitro transcription using mMESSAGE mMACHINE T7
Transcription Kit (Invitrogen, AM1344), followed by poly(A) tailing
using Poly(A) Tailing Kit (Invitrogen, AM1350) following the manu-
facturer's instructions. mRNAs were recovered by RNA Clean XP beads
(Beckman, A63987), and 50 ng/μl *Usp17le* mRNA was injected into PN3
zygotes. Embryos injected with water were used as the control for both
KD and OE experiments. All injections were performed with an
Eppendorf Transferman NK2 micromanipulator.

## CUT&Tag library preparation and sequencing
CUT&Tag was performed as previously described[72], with minor mod-
ifications. Briefly, 20000–50000 cells per sample replicate were
washed in Wash Buffer [1 mL 1 M HEPES pH 7.5 (Sigma- Aldrich, H3375),
1.5 mL 5 M NaCl (Sigma-Aldrich, S5886-1KG), 12.5 μL 2 M Spermidine
(MCE, HY-B1776), Roche Complete Protease Inhibitor EDTA-Free tablet
(Sigma- Aldrich, 4693132001), and the final volume brought to 50 mL
with dH2O], and then immobilized on 10 μL of Concanavalin A-coated
beads (Bangs Laboratories, BP531). Cells were cleared on a magnetic
rack and then permeabilized with cold antibody buffer [20 mM HEPES
pH 7.5, 150 mM NaCl, 2 mM EDTA, 0.1% BSA, 0.5 mM spermidine and 1×
protease inhibitor cocktail containing 0.05% digitonin (AbMole,
M5020-100 mg)] on ice. The cells were then incubated with primary
antibody [Anti-H3K27me3 (Millipore, 07–449, 1:100 dilution) and anti-
H2AK119ub1 (Cell Signaling Technology, 8240S, 1:100 dilution)] at
room temperature for 2 h on a shaker. Goat anti-rabbit IgG secondary
antibody (Vazyme, Ab206-01-AC) was diluted at a ratio of 1:100 in Dig-
wash buffer (20 mM HEPES pH 7.5, 150 mM NaCl, 0.5 mM spermidine,
and 1× protease inhibitor cocktail containing 0.05% digitonin) and
incubated at RT for 1 h. Cells were cleared on a magnetic rack and
washed three times with 700 μl of Dig-wash buffer. A 1:100 diluted pA-
Tn5 adapter complex combining adapter primers (Supplementary
Table 4) and pA-Tn5 according to the manufacturer's instructions
(Vazyme, S603-01) was prepared in Dig-300 Buffer [20 mM HEPES pH
7.5, 300 mM NaCl, 0.5 mM spermidine and 1× protease inhibitor
cocktail containing 0.05% digitonin]. Cells were cleared on a magnetic
rack and incubated in 100 μl of pA-Tn5 at RT for 1 h. Cells were washed
with 700 μl of Dig-300 buffer, resuspended in 300 μl of Tagmentation
buffer [10 mM MgCl2 in Dig-300 Buffer], and incubated at 37 °C for 1 h.
10 μl of 0.5 M EDTA, 3 μl of 10% SDS, and 2.5 μl of 20 mg/mL proteinase
*K* were added to each reaction to stop the tagmentation at 37 °C
overnight. DNA was purified using phenol/chloroform/isoamyl alcohol

(PCI) extraction followed by chloroform extraction and precipitated
with glycogen and ethanol. DNA was pelleted with a high-speed spin at
4 °C, washed, air dried for 5 min and resuspended in 25 μl of double-
distilled water (ddH2O) containing 100 μg/ml RNase. The DNA was
then PCR amplified using the TruePrep Index Kit V4 for Illumina
(Vazyme, TD204) and cleaned up with VAHTS DNA Clean Beads
(Vazyme, N411-01). The library quality was assessed on the Agilent
Bioanalyzer 2100 system, and the libraries were sequenced on an
Illumina HiSeq platform.

## STAR ChIP-seq library preparation and sequencing
STAR ChIP-seq was performed as previously described[62]. Briefly,
samples were lysed and S2 lysate was added as spike-in, followed by
chromatin fragmentation by micrococcal nuclease (MNase, Sigma-
Aldrich) at 37 °C for 5 min. The reaction was then terminated and the
supernatant was incubated with 1 μg of primary antibody (H3K9me3,
Active motif, #39161) overnight at 4 °C. The next day, 100 μg of Protein
A/G dynabeads (mixed at 1:1, Thermo Fisher Scientific) was added to
each sample and incubated at 4 °C. Three hours later, the beads were
washed five times with 150 μl of RIPA buffer and once with 150 μl of
LiCl buffer. For each sample, beads were resuspended with 28 μl of
H2O, 1 μl of 10× Ex Taq buffer (Takara, RR006B), and 1 μl of proteinase
K (NEB, P8107S) and incubated at 55 °C for 90 min, followed by incu-
bation at 72 °C for 40 min to inactivate the proteinase *K*. Samples were
then subjected to TruSeq library preparation using NEBNext Ultra II
DNA Library Prep Kit for Illumina (NEB, E7645S). Libraries were
sequenced using the DNBSEQ-T7 platform according to the manu-
facturer's protocol.

## Stacc-seq library preparation and sequencing
Stacc-seq was performed as previously described[4] with modifications.
Briefly, DB-1 buffer was prepared by adding 2 μl of fully dissolved 5%
digitonin (Sigma, D141) to 1 ml Buffer1. Then, 0.5 μl of pA-Tn5 and
0.5 μg of antibody (H2AK119ub1, CST, 8240S) were mixed with 7 μl DB-
1 buffer, followed by incubation at 4 °C for 30 min. Each sample was
incubated with Concanavalin-coated magnetic beads (Polyscience,
86057) for 10 min at room temperature. After removing the super-
natant, samples were then resuspended with 50 μl DB-1 buffer and 8 μl
preincubated antibody-pA-Tn5 mixture. Following incubation at 4 °C
for 2 h, the samples were washed twice with 200 μl DB-1 buffer. Each
sample was then resuspended with 50 μl DB-1 buffer and 12.5 μl 5×
TTBL (Vazyme Biotech, TD502), and the sample was incubated at 37 °C
for 30 min. Spike-in DNA added DNA purification and PCR were then
performed following the Stacc-seq protocol[4]. Sequencing was done
using the DNBSEQ-T7 platforms according to the manufacturer's
protocol.

## CUT&RUN library preparation and sequencing
CUT&RUN was performed as previously reported[64] with modifications.
Briefly, 2-cell embryos were incubated with Concanavalin-coated
magnetic beads (Polyscience, 86057) for 10 min at room tempera-
ture. Samples were then incubated with primary antibody (H3K27me3,
CST, 9733S) at a ratio of 1:100 overnight at 4 °C. The next day, after

washing for one time, beads were incubated with pA-MNase (in house, to a final concentration of 400–700 ng/ml) at 4 °C for 3 h. After washing two times, targeted digestion was performed by adding 2 μl of 100 mM $CaCl_2$ for 30 min on ice, followed by termination by adding an equal volume of 2× stop buffer supplemented with spike-in DNA. Samples were then incubated at 37 °C for 30 min for fragment releasing. The samples were then digested with proteinase K (NEB, P8107S) and purified using phenol:chloroform:isoamyl alcohol (25:24:1, v/v) followed by ethanol precipitation at −80 °C overnight. The next day, DNA was purified and subjected to TruSeq library preparation using the NEBNext Ultra II DNA Library Prep Kit for Illumina (NEB, E7645S). Sequencing was done using the DNBSEQ-T7 platforms according to the manufacturer's protocol.

## CUT&Tag, STAR ChIP-seq, Stacc-seq, and CUT&RUN data processing

CUT&Tag, Stacc, and CUT&RUN reads were aligned to the mouse mm10 genome using bowtie2[73] with the default parameters. Adaptors and aligned reads with low quality, and PCR duplicates were removed. Multi-mapping reads were included when analysing the *Dux* region. Bam files were obtained using SAMtools (v.1.9). Read coverages were estimated using bamCoverage from deepTools[74] and were visualized by IGV or UCSC genome browser.

## RNA-seq library preparation and data analysis

Poly-T oligo-attached magnetic beads were used to purify mRNA. In NEBNext First-strand Synthesis Reaction Buffer (5×), RNA was fragmented at high temperature using divalent cations. The segmented mRNA was used as a template, and first strand cDNA was synthesized using random hexamer primers and M-MLV reverse transcriptase (RNase H). The second strand of cDNA was then synthesized by adding dNTPs, DNA polymerase I, and RNase. The 3′ end was then adenylated. The NEB Next Adaptors connector was connected and ready for hybridization. The AMPure XP system (Beckman Coulter, Beverly, USA) was used to purify the library fragments. The target fragment was then treated with 3 μl of USER enzyme (NEB, USA) at 37 °C for 15 min, followed by 5 min at 95 °C, and PCR was performed. PCR products were purified using the AMPure XP system. After quality assessment, the library was sequenced on an Illumina HiSeq platform. Raw reads were processed using trim-galore, and clean reads were mapped to mm10 by Hisat2 with default parametes. Uniquely mapped reads were calculated by FeatureCounts, with default parameters. Raw counts were normalized by library size via transcripts per kilobase of exon model per million mapped reads (TPM). Differentially expressed genes (DEGs) (Fold change > 2, $P < 0.05$) between different groups were analysed using Deseq2.

## Statistics and reproducibility

For qPCR, statistical tests were performed using data from three biological replicates using a two-tailed unpaired *t*-test. For imaging analysis, a two-tailed unpaired *t*-test was performed with data from at least three samples, and ANOVA was used to compare more than two groups and expressed as the mean ± SEM. *P* values less than 0.05 were considered significant (\*$p < 0.05$, \*\*$p < 0.01$ or \*\*\*$p < 0.001$). Western blot analysis was performed by densitometry using ImageJ software (RRID:SCR_003070). Statistical analyses for significance used a two-tailed unpaired *t*-test. \*$p < 0.05$, \*\*$p < 0.01$, \*\*\*$p < 0.001$, \*\*\*\*$p < 0.0001$ were considered to be statistically significant. Error bars represent the standard deviation of the mean in three independent experiments.

## Reporting summary

Further information on research design is available in the Nature Portfolio Reporting Summary linked to this article.

## Data availability

All sequencing data generated in this study have been deposited to NCBI GEO. The accession numbers for RNA-seq data are GSE223067 (*Usp17l* KD mESCs) [https://www.ncbi.nlm.nih.gov/geo/query/acc.cgi?acc=GSE223067], GSE234366 (*Usp17le* OE mESCs). [https://www.ncbi.nlm.nih.gov/geo/query/acc.cgi?acc=GSE234366] GSE277982 (WT/*Dux* KD/*Dux* KD-*Usp17le* OE/*Usp17l* OE/ *Usp17l* OE-*Dux* KD mESCs). [https://www.ncbi.nlm.nih.gov/geo/query/acc.cgi?acc=GSE277982], and GSE282420 (*Usp17l* KD and *Usp17le* OE late 2C embryos) [https://www.ncbi.nlm.nih.gov/geo/query/acc.cgi?acc=GSE282420]. The accession number for CUT&Tag data is GSE224711. The accession number for STAR ChIP-seq, Stacc-seq, and CUT&RUN data is GSE282421. Source data are provided with this paper.

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

## Acknowledgements

This study was supported by funding from the National Key Research and Development Program of China (2018YFA0107000 to L.Liu, 2021YFA1100102 to W.X.), the National Natural Science Foundation of China (32030033, 82230052, and 32261160571 to L.Liu), Haihe Laboratory of Cell Ecosystem Innovation Fund (22HHXBSS00029 to L.Liu), and the Tsinghua-Peking Center for Life Sciences (W.X.). W.X. is a New Cornerstone Investigator. We acknowledge Tianjin Novogene Technology for their high-throughput sequencing services. We thank Haifeng Fu, Weiyu Zhang, Niannian Li, and Jiyu Chen for assisting the experiments.

## Author contributions

P.S. performed most of the experiments in mouse ESCs and data analysis and drafted the manuscript; X.L. performed the experiments in mouse embryos with the help of W.W., L.W. and L.D., analyzed and interpreted the data, and revised the manuscript; K.J. participated in telomere measurement, flow cytometry, construction of the CUT&Tag library, immunofluorescence experiments and microscopic imaging. L.L.Liu and G.Y. performed bioinformatics analyses; J.Y. was involved in some of the ubiquitination experiments. W.X. advised the project and bioinformatics analysis, discussed the results, and revised the manuscript; L.Liu planned the project, designed the experiments, supervised data analysis, and revised the manuscript.

## Competing interests

The authors declare no competing interests.
