## [Peer Review file · Nature Communications]

USP17L promotes the 2-cell-like program through deubiquitination of H2AK119ub1 and ZSCAN4

Corresponding Author: Professor Wei Xie

Version 0:

Reviewer comments:

Reviewer #1

(Remarks to the Author)

USP17L is transiently activated at ZGA and 2CLC ESCs. In this manuscript, the authors investigated its function in ESCs and found that USP17L deubiquitinates not only H2AK119ub1 at the DUX locus to promote DUX expression, but also ZSCAN4 protein to prevent its proteasome-mediated degradation. Both functions may contribute to promotion of 2CLC state. These findings are of interest to the field of pluripotent stem cells, epigenetics, and development. The manuscript is clearly written, and the data are well presented. Meanwhile, I have several concerns.

Major points

1. Because this study only studies ESCs, the relevance to real 2-cell embryos remains unknown. This is important when considering the current topic of the 2CLC field. Regardless of positive or negative outputs, it is necessary to include the data of whether H2AK119ub1 level at the DUX locus is correlated with DUX expression in embryos and how much Usp17l expression is defective in DUX KO embryos. Re-analyses of public datasets will address these questions.
2. The last sentence in abstract says USP17L is a "critical regulator" of the 2C state. I feel this is overstating because its KD reduces Zscan4-positive cells by only half (Fig 2E). If a factor is indeed "critical", one should assume that KD/KO would reduce the 2CLC population to nearly zero. I recommend tuning down the statement.
3. Fig 2. How USP17L suppress pluripotent genes in ESCs? Is it related to deubiquitination activity? Does USP17L have other functions for pluripotent gene regulation? Is it known that KD/KO of other 2C genes similarly cause upregulation of pluripotent genes in ESCs? Discussion is required.
4. Fig 4. Analysis of H2AK119ub1 and H3K27me3 in Usp17l KD ESCs are not sufficient. While I admit that the Dux locus is the focus of this study, more comprehensive analyses of the data are required. For instance, how many genes (and to what extent) these two marks increase and decrease upon KD? What is the rank order of the level of change at Dux in all other genes differentially modified upon KD? What are functional and genomic features of the differential genes/peaks/regions? Such analyses would provide additional insights into the function and the chromatin targeting mechanisms of UPS17L, which is lacking in the current manuscript.

Minor points

Line 72.

It is fair to also cite De Iaco et al., 2020 (PMID 31806660)

Introduction.

The authors may want to cite Baek et al., 2012 (PMID: 22984479) that claims the lethality of DUB2 KO embryos.

Fig.1A

There are two forms of Usp17b and three forms of Usp17e. Are those splicing isoforms? Please describe in the legend. Also include the information in somewhere about which isoforms used in overexpression experiments.

Fig.3F

What are the three groups?

Experimental procedures

The method of flow cytometry is not found.

Reviewer #2

(Remarks to the Author)

Zygotic genome activation (ZGA) is a pivotal event during the maternal-to-zygotic transition and early embryo development. 2-cell-like cells (2CLCs) mimic the characteristics of 2-cell embryos, including similar transcriptomic profiles, epigenetic features, and developmental potential. Consequently, they serve as a valuable model for investigating ZGA and totipotency. In this study, techniques such as knockdown, overexpression, and biochemical assays were employed in 2CLCs to uncover the regulatory role of USP17L in promoting the 2-cell-like state. The findings indicate that USP17LE may modulate the expression of Dux through H2AK119ub1 and H3K27me3 modifications. While this discovery is intriguing, further experiments are necessary to validate this conclusion. Detailed comments on the study are provided below:

Major:

1. As the current findings mainly rely on mESCs, it is crucial to clarify whether USP17L also regulates DUX in mouse fertilized embryos. Specifically, the impact of Usp17l Knockdown (KD) and overexpression (OE) on the expression level of Dux and the epigenetic marks around Dux locus (e.g., H2AK119ub1 and H3K27me3) in embryos should be investigated.
2. Figure 3C - Various histone modifications were analyzed in Usp17l KD ESCs using western blot. Considering that H2BK120ub1 has been reported to stimulate H3K4me3 deposition in human cells and during ESC differentiation (Kim et al., Cell, 2009.; Chen et al., Cell Research, 2012.), it is necessary to examine the effects of Usp17l KD on H2BK120ub1 and H3K4me3 levels in ESCs.
3. Figure 3C - The mechanism underlying how Usp17l KD results in a substantial reduction of H3K9me2/3 should be investigated. ChIP-seq of H3K9me2/3 should be analyzed in Usp17l KD ESCs. One possibility is that Usp17l KD attenuates the expression of ZGA genes, including Zfps, which are associated with the establishment of H3K9me2/3.
4. Figure 4A - It would be valuable to determine whether Usp17le OE in Dux KD ESCs leads to increased expression levels of Dux, Usp17l, Zscan4, and other ZGA genes. Additionally, performing mRNA-seq analysis in addition to qPCR for selective genes would enhance the comprehensiveness of the results.
5. Figure 5 - The authors proposed that USP17L can stabilize ZSCAN4 through direct deubiquitination. Whether USP17L can also reduce H2AK119ub1 levels to derepress Zscan4, like Dux? Examining H2AK119ub1 and H3K27me3 signals around Zscan4 loci and the expression level of Zscan4 in Usp17l KD ESCs could provide further insights into the regulatory mechanisms involved.

Minor:

1. Line 202 - "Consistent with or hypothesis" may refer to "Consistent with our hypothesis".
2. Figure 4C, D - The TPM values of Dux RNA in the two Control groups differed widely. Including a few more replicates would be beneficial to eliminate the batch effect.

Reviewer #3

(Remarks to the Author)

The authors present a very interesting study showing that USP17 proteins functionally regulate mouse embryonic stem cell 2C-like state through deubiquitinating H2A (promoting Dux expression) and ZSCAN4 (preventing its proteasomal degradation). The manuscript is well written and the experiments are logically designed, although more information in the Methods section is required (as outlined below). I have no major concerns.

I have the following minor suggestions and comments:

1. The title of the paper is a little lacking – "USP17L promotes 2-cell-like state through deubiquitination of H2A and ZSCAN4" or something similar might be more informative.
2. Given the genetic complexity of the USP17-like family, it may be helpful to also refer to the other/previous names assigned to murine family members when they are first introduced in the Results section e.g. Usp17la (Dub1), Usp17lb (Dub1a) Usp17lc (Dub2), Usp17ld (Dub2a), Usp17le (Dub3).
3. There is some key relevant USP17 literature that at the very least needs citing, and could be discussed in the context of the authors' data. Esrrb has been shown to upregulate USP17 expression in mESCs (which is also influenced by coactivator NCoA1 - 10.1016/j.molcel.2013.10.003, journal.pone.0093663). This is interesting as Esrrb is found to be upregulated following USP17 depletion in this manuscript. Furthermore, histone deubiquitination by USP17 has also previously been reported in human cells (H2AX), which was found to be important in maintaining genome integrity through regulation of DDR (10.1016/j.molonc.2014.03.003).
4. There seem to be antibodies missing from the Key Resources Table – most notably the USP17 antibody.
5. There appears to be no information in the Methods section on which overexpression or knockdown plasmids were used in

the study.

6. In the Methods section "Mutation of USP17LE", while the mutants are defined, there should be information on how the mutagenesis was carried out here – was a commercial kit used?
7. There are two cell lines used in the study, but there appears to be very little information on which cell line was used for which experiments in either the Methods section or the Figure legends.
8. In Figure 3B it is unclear where these sequences have been derived from – is USP17L supposed to represent USP17LE? Some protein identifiers in the legend would be helpful.
9. In Figures 4C, 5F and 5K, r1 and r2 should be defined – presumably replicate 1 and 2?
10. In Figure 5E, it is worth noting that USP17LE overexpression appears to have a global effect on ubiquitination (based on Input lanes). This has previously been reported with overexpression of human USP17 (10.1038/cr.2010.41), showing that murine family members are also very active enzymes and there are likely to be global cellular impacts from ectopic expression of USP17LE.

Version 1:

Reviewer comments:

Reviewer #1

(Remarks to the Author)

The revised manuscript has satisfactorily addressed all my comments. This is a very nice work that, for the first time, demonstrates the link between H2Aub1 and 2C program. Although it seemed that the KD experiment in early embryos unfortunately gave negative data, I am sure that these data will also be highly appreciated in the field.

A few minor comments to be corrected before publication:
Fig 6E. Please quantify H2Aub1 and H3K27me3 signals.

Fig.5, 6E. The scales of ChIP-seq signals are wanted, if possible.

Fig.6, 7. In Usp171 KD or OE experiment, please mention what their respective controls are? (Are they non-injected embryos?)

Line 239-285 should be shortened.

Line 361-379 should be moved to discussion.

Congratulations,
Azusa Inoue

Reviewer #3

(Remarks to the Author)

The authors have addressed all of my comments in the revised manuscript.

Reviewer #4

(Remarks to the Author)

Based on the review of the manuscript and the rebuttal letter, particularly the response to Reviewer 2, I believe that the authors still have considerable work to do in addressing the reviewers' concerns. The authors have performed additional analyses using publicly available datasets and experimental validations to clarify the role of USP17L in regulating DUX and epigenetic modifications in mouse embryos, as well as other related analyses.

However, the new study, instead of supporting a definitive role for USP17L in mouse early embryos, underscores the contrasting effects of USP17 knockdown in embryonic stem cells (ESCs) versus embryos. Specifically, USP17L appears to regulate 2C-like gene expression and developmental programs differently in these two contexts. In ESCs, USP17L knockdown led to a downregulation of 2C genes, such as Zscan4, Dux, and retrotransposons, while upregulating pluripotency genes like Nanog and Oct4. In contrast, in embryos, USP17L knockdown had a much milder effect on H2AK119ub1 and did not significantly impact pre-implantation development or major ZGA, which contradicts the claims made in the manuscript. This discrepancy is particularly relevant to the manuscript's title and its major claims. Additionally, overexpression of USP17L in embryos may artificially reduce H2AK119ub1 and activate 2C genes, further raising concerns about the physiological relevance of these findings. These divergent effects question the extent to which conclusions drawn from ESC models are applicable to actual embryonic development.

Beyond this major concern, further clarity is needed on the details of the immunoprecipitation experiments, especially those involving histone modifications. It is important to note that histones do not exist individually in solutions but as part of nucleosomes, which are not typically released under routine lysis conditions. Therefore, the methods used to extract and study histone modifications need to be described in detail to ensure the validity and reproducibility of the findings.

Reviewer #5

(Remarks to the Author)

The manuscript "USP17L promotes the 2-cell program through deubiquitination of H2AK119ub1 and ZSCAN4" describes the functional role of Usp17l during minor ZGA by deubiquitinating H2AK119ub1 and promoting the expression of Dux, and deubiquitinating and stabilizing ZSCAN4. I have not evaluated this manuscript in the first round, but I feel that the authors address the comments made during the initial evaluation very well. Also, I feel that the authors present a wide range of assays broadly supporting their conclusions.

However, I do think that the proteomics part of the manuscript requires significant improvement to become meaningful:

- It is very worrying that the authors do not find H3 in Figure 2D in the "USP17L" condition, as this is part of the core histone to which also H2A belongs. In particular since H3 clearly shows up in Fig 2A as binding to USP17L. Also, given the strong co-localisation of H3K27me3 and H2AK119ub, it is remarkable that there are no H3K27me3 signals in the "USP17L" condition.

- The LC-MS results of the USP17L co-IP are very poorly analyzed and cannot be interpreted or evaluated with the current analysis (Fig 2a). The authors should properly analyze the MS interaction results showing a volcano plot with showing log2 fold changes over control (x-axis), and p-values (y-axis). What are the proteins that the authors find significantly interacting with USP17L?; please label these in such graph. Also, only such analysis will reveal whether indeed H2A, H3, and H4 histones are enriched as compared to background. Finally, these should show Zscan4 as interactor in this analysis, which is what the authors later claim in the manuscript.

- There is no description in the materials and methods of the MS experiments and their analysis.

- Please put an asterisk at USP17L on the SDS-PAGE of Fig 2a, so that readers can appreciate this positive control of the pulldown. Similarly, in Fig 2e, please point in the graph where to look for the relevant differences between wt and mutant overexpression (I understood in the end, but pointing to these would help). Btw. Fig 2e does not seem to be an in vitro assay, as outlined in the text, but an in vivo assay?

- The HA pulldown with flag-read-out, Fig 4e, is non-convincing, with a very faint band. ThUSP17LE-Zscan4 interaction should therefore be confirmed by alternative means, for which the authors can conveniently use their MS results as outlined above. If Zscan4 does not show up as interact in the MS results, further confirmations are even more essential.

Version 2:

Reviewer comments:

Reviewer #4

(Remarks to the Author)

The author has responded to my comments in a satisfactory and thorough manner. In my opinion, the manuscript now meets the standards for publication.

Reviewer #5

(Remarks to the Author)

By removing the interaction proteomics from their paper, the authors have addressed my main concern. While I feel removing this data affects the comprehensiveness of their study, I agree with the authors that it does not directly affect the main conclusions of the paper. Also my other concern has been addressed by performing the reciprocal pulldown.

Based on the comments of reviewer #4, I do think it would be useful if the authors additionally include a part in the discussion where they outline their view on the role of USP17L during early embryogenesis. Because USP17L can reprogram ES cells towards the 2C like state, but such event is unlikely to happen in vivo. So also taking into account the differences between the ES cells and in vivo embryos in their finding on USP17L, it would be interesting if the authors better indicate what role they actually envision for USP17L in early embryogenesis, and at what stage.

AUTHORS' RESPONSES TO REVIEW

We deeply appreciate the reviewers for the constructive and valuable comments and suggestions, which have led to significant improvement of the manuscript. Here we include a letter to address the issues raised in the previous submission. Please note that we used Fig. 1, 2, 3, etc. to refer to figures in the revised manuscript and Fig. R1, R2, R3, etc. to refer to figures in this letter.

For the reviewers' convenience, we also included a version of the manuscript in which the revised sections related to our responses to the reviewers' comments are marked with blue.

POINT-BY-POINT RESPONSES

Reviewer #1:

USP17L is transiently activated at ZGA and 2CLC ESCs. In this manuscript, the authors investigated its function in ESCs and found that USP17L deubiquitinates not only H2AK119ub1 at the DUX locus to promote DUX expression, but also ZSCAN4 protein to prevent its proteasome-mediated degradation. Both functions may contribute to promotion of 2CLC state. These findings are of interest to the field of pluripotent stem cells, epigenetics, and development. The manuscript is clearly written, and the data are well presented.

Response: We really appreciate these positive and encouraging comments.

1. Because this study only studies ESCs, the relevance to real 2-cell embryos remains unknown. This is important when considering the current topic of the 2CLC field. Regardless of positive or negative outputs, it is necessary to include the data of whether H2AK119ub1 level at the DUX locus is correlated with DUX expression in embryos and how much Usp17l expression is defective in DUX KO embryos. Re-analyses of public datasets will address these questions.

Response: Thank you for these constructive suggestions. We fully agree that data from embryos are critical to consolidating the conclusion. Regarding the first question, by reanalyzing a publicly available dataset (Mei et al., 2021), we found that H2AK119ub1 was highly enriched near *Dux* in mouse oocytes, decreased at the 1-cell and early 2-cell stages when *Dux* was activated (Fig. R1A-C) (Xiong et al., 2022; Zhang et al., 2016), and then increased and remained at high levels at the late 2-cell stage until the blastocyst stage (Fig. R1A). Therefore, the dynamics of H2AK119ub1 at the *Dux* locus negatively correlated with the expression changes of *Dux* during mouse pre-implantation development. We also analyzed H3K27me3 at *Dux* and found that H3K27me3 was also enriched in mouse oocytes, decreased after fertilization, but was not restored until in ICM (Fig. R1D). The slower restoration of H3K27me3 was consistent with the lack of de novo H3K27me3 at promoters in mouse pre-implantation embryos (Zheng et al., 2016). These data also indicate that H3K27me3 may be not a major repressor of *Dux* in mouse pre-implantation embryos. Please note that as the *Dux* locus exists as tandem repeats (Grow et al., 2021), multi-mapping reads were included when studying chromatin marks in this region. Although the actual enrichment levels were unknown due to the repetitive nature of this region, the collective signals should reflect the overall enrichment at the *Dux* locus and it should

not affect the comparison between different samples and stages. In fact, such result was validated by the strong H3K27me3 in mESCs but low H3K27me3 in mouse embryos at the *Dux* locus (Fig. R1D), as well as the strong H2AK119ub1 in control mouse embryos but low H2AK119ub upon *Usp17le* OE in mouse embryos (Fig. R5A, “OE” in “H2AK119ub1” at the *Dux* locus, discussed later, page 10 in the letter).

Regarding the second question, in *Dux* KO late 2C embryos, *Usp17l* expression was partially compromised as analyzed using a published dataset (Fig. R1E) (Chen and Zhang, 2019). The partial reduction was consistent with the notion that *Dux* KO only had a mild effect on ZGA and mouse early development (Chen and Zhang, 2019; De Iaco et al., 2020; Guo et al., 2019).

Figure R1. H2AK119ub1/H3K27me3 at *Dux* locus, *Dux* expression and the effect of *Dux* KO on *Usp17l* expression in mouse early embryos. (A) The UCSC genome browser view showing global H2AK119ub1 (top) and that near *Dux* locus (bottom) in mouse early embryos. H2AK119ub1 data are from published data (Mei et al., 2021). FGO, full-grown oocytes; 1C, 1-cell; E2C, early 2-cell, L2C, late 2-cell; Mo, morula; BI, blastocyst. H2AK119ub1 in mouse ESCs is shown as a comparison. (B-C) Line charts showing the expression (RPKM) of *Dux* during mouse pre-implantation development at both the transcriptional (B) and translational (C) levels. RNA-seq and Ribo-seq data are from (Zhang et al., 2016) and (Xiong et al., 2022), respectively. (D) The UCSC genome browser view showing global H3K27me3 (top) and that near *Dux* locus (bottom) in mouse early embryos. H3K27me3 data are from published data (Zheng et al., 2016). FGO, full-grown oocytes; 1C, 1-cell; E2C, early 2-cell, L2C, late 2-cell; 8C, 8-cell; ICM, inner cell mass. H3K27me3 in mESC is shown as a comparison. (E) Bar charts showing the expression (TPM) of *Usp17l* in wild-type (WT) and *Dux* knockout (KO) mouse 2-cell embryos. RNA-seq data are from published data (Chen and Zhang, 2019).

2. The last sentence in abstract says USP17L is a “critical regulator” of the 2C state. I feel this is overstating because its KD reduces Zscan4-positive cells by only half (Fig 2E). If a factor is indeed “critical”, one should assume that KD/KO would reduce the 2CLC population to nearly zero. I recommend tuning down the statement.

Response: Thank you for this suggestion. We agree and have toned down the statement and changed the "critical regulator" to "key regulator", and included the new statement in the revised manuscript (line 47, page 2).

3. Fig 2. How USP17L suppress pluripotency genes in ESCs? Is it related to deubiquitination activity? Does USP17L have other functions for pluripotency gene regulation? Is it known that KD/KO of other 2C genes similarly cause upregulation of pluripotent genes in ESCs? Discussion is required.

Response: We appreciate these great questions. Acquisition of totipotency or expanded developmental potential is often accompanied by downregulation of pluripotency genes for reasons that are not very clear (Genet and Torres-Padilla, 2020; Lu and Zhang, 2015). A similar observation was also made in our work. How USP17L suppresses pluripotency genes in mESCs remains unclear. The IP-MS and IP experiments did not show interactions between USP17L and pluripotency-associated proteins (Table S1 in the revised manuscript), indicating that USP17L may not deubiquitinate these proteins directly. We cannot exclude the possibility that USP17L promotes the expression of or stabilizes the negative regulators of pluripotency genes through deubiquitination, which warrants further investigation. We have discussed this in the revised manuscript (lines 134-140, page 5).

We reanalyzed publicly available data (Huang et al., 2021; Zhang et al., 2019), which showed that *Dux* knockout (KO) (Fig. R2A) or *Zscan4* knockdown (KD) (Fig. R2B) in mESCs also caused the upregulation of pluripotency genes, concomitant with the downregulation of 2C genes.

Figure R2. The expression of pluripotency genes in *Dux* knockout (KO) or *Zscan4* knockdown (KD) mESCs. (A-B) Heatmap showing the expression levels and fold changes of pluripotency genes and representative 2C genes in *Dux* KO mESCs (A) and *Zscan4* KD ESCs (B). Both the raw data (left in each panel) and fold change (right in each panel) are shown. RNA-seq data from previous studies (Huang et al., 2021;

Zhang et al., 2019) were analyzed.

4. Fig 4. Analysis of H2AK119ub1 and H3K27me3 in *Usp17l* KD ESCs are not sufficient. While I admit that the *Dux* locus is the focus of this study, more comprehensive analyses of the data are required. For instance, how many genes (and to what extent) these two marks increase and decrease upon KD? What is the rank order of the level of change at *Dux* in all other genes differentially modified upon KD? What are functional and genomic features of the differential genes/peaks/regions? Such analyses would provide additional insights into the function and the chromatin targeting mechanisms of *Usp17l*, which is lacking in the current manuscript.

Response: Thank you very much for these great suggestions, according to which we have reanalyzed our data. We calculated the enrichment of H2AK119ub1 and H3K27me3 at gene promoters in mESCs upon *USP17L* KD. Both H2AK119ub1 and H3K27me3 were globally increased (Fig. R3A-B), consistent with the western blot result (Fig. 2C in the revised manuscript). Further analyses showed that H2AK119ub1 increased at a large number of gene promoters (n=3149) (fold change of RPKM ≥ 1.5) in *Usp17l* KD mESCs, with only a decrease near a limited number of genes (n=30) (Fig. R3C, left). A similar trend, but to a lesser extent, was observed for H3K27me3 (increased, n=1930; decreased, n=1366) (Fig. R3C, right). Notably, the increase of H2AK119ub1 (Fig. R3C, left) and H3K27me3 (Fig. R3C, right) near *Dux* ranked number 7 and number 1, respectively, among all the genes. Given that *Dux* can also activate *Usp17l* (De Iaco et al., 2017; Hendrickson et al., 2017), the possible positive feedback loop between *Dux* and *Usp17l* may enhance the changes of H2AK119ub1 and H3K27me3 at the *Dux* locus. Among minor ZGA genes, only *Dux* was strongly enriched for H2AK119ub1 and H3K27me3 (Fig. R3C, red), raising a possibility that the downregulation of other minor ZGA genes upon *Usp17l* KD may be regulated indirectly.

Genes exhibiting an increase of promoter H2AK119ub1 were mainly involved in development and cell differentiation (Fig. R3C, bottom left). This was also true for those with increased promoter H3K27me3 (Fig. R3C, bottom right), consistent with the functions of H2AK119ub1 and H3K27me3 in regulating developmental genes (Blackledge and Klose, 2021). Genes with decreased H2AK119ub1 were not enriched in any GO terms, likely due to the small number of genes (n=30). Genes with decreased H3K27me3 at gene promoters upon *Usp17l* KD were involved in protein transport, spermatogenesis, DNA damage response, and apoptosis (Fig. R3C, bottom right). The increase of H2AK119ub1 and H3K27me3 was not associated with gene expression changes globally (Fig. R3D, grey, $R=-0.06$ to -0.02), including those at developmental genes (Fig. R3D, black), consistent with the fact that these genes are already repressed in WT ESCs. These data suggest that while H2AK119ub1 and H3K27me3 were globally altered in *Usp17l* KD mESCs, the changes at the *Dux* locus are among the most pronounced ones. We have incorporated these new data into our revised manuscript (Fig. 3E-F and S4 in the revised manuscript, pages 40 and 54).

Figure R3. The effect of USP17L KD on H2AK119ub1 and H3K27me3. (A) Line charts showing the global H2AK119ub1 (left) and H3K27me3 (right) in WT and USP17L KD mESCs. (B) The UCSC genome browser view showing the global H2AK119ub1 and H3K27me3 and that at the *Dux* locus in WT and USP17L KD mESCs. (C) Scatter plots showing H2AK119ub1 (left) and H3K27me3 (right) at gene promoters in WT and USP17L KD mESCs. The *Dux* locus is shown. Note its value is high as *Dux* is a multi-copy gene, and multi-mapping reads were used. The purple dashed line indicates the RPKM values when they show equal values between control and *Usp17l* KD mESCs. The black dashed line indicates H2AK119ub1/H3K27me3 changes at gene promoters when they equal to that at the *Dux* locus upon USP17L KD. The Gene Ontology terms of genes exhibiting increased/decreased H2AK119ub1 and H3K27me3 are shown at the bottom. (D) Scatter plots comparing gene expression changes and the changes of H2AK119ub1 (left) and H3K27me3 (right) at gene promoters. Developmental genes and minor ZGA genes are highlighted in black and red, respectively.

Minor points

Line 72.

It is fair to also cite De Iaco et al., 2020 (PMID 31806660)

Response: Thank you for the suggestion. We have included this reference in the revised manuscript (line 69, page 3 in the revised manuscript).

Introduction.

The authors may want to cite Baek et al., 2012 (PMID: 22984479) that claims the lethality of DUB2 KO embryos.

Response: Thank you for the suggestion. We have included this reference in the revised manuscript (line 87, page 4 in the revised manuscript).

Fig.1A.

There are two forms of *Usp17lb* and three forms of *Usp17le*. Are those splicing isoforms? Please describe in the legend. Also include the information in somewhere about which isoforms used in overexpression experiments.

Response: The two forms of *Usp17lb* and three forms of *Usp17le* are different splicing isoforms. We have now described this in the figure legend (lines 1028-1030, pages 35-36 in the revised manuscript). The *Usp17le* and *Usp17lb* isoforms used in this paper are both isoform 1 as shown in Figure 1A. These isoforms contain the complete deubiquitination domain. We have now included the information in EXPERIMENTAL MODEL AND SUBJECT DETAILS (lines 502-504, page 20 in the revised manuscript).

Fig.3F

What are the three groups?

Response: We apologize for the missing information. These three groups refer to mESCs with different USP17LE fluorescence intensities (green), indicating different *Usp17le* overexpression levels. Higher levels of USP17LE caused more decrease of H2AK119ub1. We have included this information in the revised figure legend (lines 1084-1085, page 38).

Experimental procedures

The method of flow cytometry is not found.

Response: Thank you for pointing this out. We have added the method of flow cytometry in the revised manuscript (lines 564-572, pages 21-22). The details are as follows:

The cells were resuspended in FACS Buffer (0.1% BSA in PBS) and filtered through a 40- μ m filter (Falcon) to remove large cell clumps, followed by analysis using a FACS Aria II flow cytometer (BD, 85 μ m nozzle). The area-scaling factor was set, and forward scatter (FSC)-A and side scatter (SSC)-A were used to exclude large-sized cell structures or debris, with SSC-W set to avoid contamination by doublets or triplets. Finally, the appropriate fluorescence channels corresponding to the excitation light (594nm) were used for cell sorting.

Reviewer #2:

Zygotic genome activation (ZGA) is a pivotal event during the maternal-to-zygotic transition and early embryo development. 2-cell-like cells (2CLCs) mimic the characteristics of 2-cell embryos, including similar transcriptomic profiles, epigenetic features, and developmental potential. Consequently, they serve as a valuable model for investigating ZGA and totipotency. In this study, techniques such as knockdown, overexpression, and biochemical assays were employed in 2CLCs to uncover the

regulatory role of USP17L in promoting the 2-cell-like state. The findings indicate that USP17LE may modulate the expression of *Dux* through H2AK119ub1 and H3K27me3 modifications. While this discovery is intriguing, further experiments are necessary to validate this conclusion.

Response: We sincerely appreciate these comments and have revised the manuscript as suggested.

1. As the current findings mainly rely on mESCs, it is crucial to clarify whether USP17L also regulates DUX in mouse fertilized embryos. Specifically, the impact of *Usp17l* Knockdown (KD) and overexpression (OE) on the expression level of *Dux* and the epigenetic marks around *Dux* locus (e.g., H2AK119ub1 and H3K27me3) in embryos should be investigated.

Response: Thank you for these comments and suggestions. We fully agree that data from the embryos will be highly valuable to elucidate the function and the underlying mechanism of USP17L *in vivo*. By reanalyzing a publicly available dataset (Mei et al., 2021), we found that H2AK119ub1 was highly enriched near *Dux* in mouse oocytes, decreased at the 1-cell and early 2-cell stages when *Dux* was activated (Fig. R1A-C) (Xiong et al., 2022; Zhang et al., 2016), and then increased and remained at high levels at the late 2-cell stage until the blastocyst stage (Fig. R1A). Therefore, the dynamics of H2AK119ub1 at the *Dux* locus negatively correlated with the expression changes of *Dux* during mouse pre-implantation development, indicating the potential regulation of *Dux* by H2AK119ub1 in mouse embryos. H3K27me3 is less likely involved in *Dux* regulation as it was not restored until in ICM after decreasing following fertilization (Fig. R1D, and Fig. 5C in the revised manuscript). Please refer to the first comment of Review #1 (page 1 in the letter).

As suggested, we injected *Usp17l* siRNAs (KD) or *Usp17le* mRNAs (OE) in mouse zygotes at pronuclear stage 3 (PN3) and examined gene expression, H2AK119ub1 and H3K27me3 at the late 2-cell stage (Fig. R4A). We confirmed the efficient *Usp17l* KD and OE in mouse late 2-cell (2C) embryos (Fig. R4B-C). *Usp17l* KD did not have an apparent impact on mouse pre-implantation development (Fig. R4D-E, compare “KD” with “Ctrl”) or major ZGA (Fig. R4F, left, and R4G). Minor ZGA genes showed only moderate downregulation in *Usp17l* KD late 2C embryos (Fig. R4F, left, and R4G), as exemplified by *Zscan4* family genes (Fig. R4H, left, “KD”). *Dux* was evidently downregulated (Fig. R4H, right, “KD”). On the other hand, *Usp17le* OE embryos developed normally to the 2C stage, but showed partial delay at the 4C stage, and finally arrested at the 8C-morula stage (Fig. R4D-E, compare “OE” with “Ctrl”). Despite the largely normal maternal RNA decay at the late 2C stage (Fig. R4F, right), minor ZGA genes, including *Usp17l* and *Zscan4* family genes and *Dux*, were upregulated (Fig. R4F, right, and R4G-H). Additionally, several classes of repeats, including LINE1 (L1Md_F, L1Md_T) and MERVL (MT2_Mm, MERVL_int) that are normally activated in minor ZGA were also upregulated upon *Usp17le* OE, although they were largely unaffected upon *Usp17l* KD (Fig. R4I). Major ZGA genes were partially decreased possibly due to slight developmental delay at the late 2C stage (Fig. R4G). It should be noted that the upregulation of minor ZGA genes was unlikely due to developmental delay, as the levels of activation were dramatically increased at levels much higher than those in control embryos at the late 2C stage (Fig. R4G) when the levels of minor ZGA peaked in WT embryos (Fig. R4J).

The moderate effect of *Usp17l* KD on H2AK119ub1 and mouse ZGA may not be too surprising. First,

USP17L may be redundant with other deubiquitinases such as USP16, a maternal protein that can deubiquitinate H2AK119ub in oocytes and is present in both oocytes and early embryos (Rong et al., 2022). Second, although *Dux* expression was evidently affected upon *Usp17l* KD, the downstream effect of *Dux* is possibly compensated by other regulators of minor ZGA, given that *Dux* KO only had a mild effect on ZGA in mouse embryos (Chen and Zhang, 2019; De Iaco et al., 2020; Guo et al., 2019). For example, OBOX has been shown to regulate many targets of DUX (Ji et al., 2023; Yang et al., 2024) and OBOX4 redundantly drives ZGA in the absence of DUX (Guo et al., 2024). Of note, *Usp16* and *Obox4* were only moderately expressed in mESCs compared to those in mouse oocytes or pre-implantation embryos (Fig. 2A in (Rong et al., 2022), and Fig. 1b in (Ji et al., 2023)), which may account for the lack of such redundancy in mESCs. Together, these results were consistent with the observations in mESCs, indicating the involvement of USP17L in minor ZGA gene regulation in mouse embryos.

Figure R4. The effect of *Usp17l* knockdown (KD) or *Usp17le* overexpression (OE) on gene expression in mouse embryos. (A) Schematic showing injection of siRNA cocktail targeting all *Usp17l* family genes or *Usp17le* mRNAs into mouse PN3 zygotes and examination of RNA, H2AK119ub1 and H3K27me3 at the late 2C stage. (B-C) Bar charts showing the transcription levels (TPM) of *Usp17l* family genes after *Usp17l* KD (B) and *Usp17le* OE (C) in mouse late 2C embryos. (D) Bar chart showing the developmental ratios of mouse control, *Usp17l* KD, and *Usp17le* OE pre-implantation embryos. The expected developmental stage of mouse embryos *in vitro* at the indicated time point are shown on the right. (E) The development of mouse control, *Usp17l* KD and *Usp17le* OE pre-implantation embryos. Scale bar, 50 μ m. (F) Scatter plots showing the changes of maternal (green), minor ZGA (blue), and major ZGA (red) genes upon *Usp17l* KD (left) or *Usp17le* OE (right) in late 2C embryos. (G) Box plot showing the expression changes of minor and major ZGA genes in *Usp17l* KD and *Usp17le* OE late 2C embryos. Global gene expression (“All”) is shown as a control. (H) Bar charts showing the expression (TPM) of *Zscan4* (left) and *Dux* (right) in *Usp17l* KD and *Usp17le* OE late 2C embryos. (I) Scatter plots showing the expression changes of repeats in *Usp17l* KD (left) and *Usp17le* OE (right) late 2C embryos. (J) Bar chart showing the expression levels of minor genes in WT embryos during mouse pre-implantation development. FGO, Full-grown GV oocyte; MII, metaphase II oocyte; 1C, 1-cell; E2C, early 2-cell; L2C, late 2-cell; 4C, 4-cell; 8C, 8-cell; ICM, inner cell mass.

We then examined H2AK119ub1 and H3K27me3 in *Usp17l* KD and *Usp17le* OE late 2C embryos. Spike-in normalized Stacc-seq for H2AK119ub1 (Liu et al., 2020) and CUT&RUN for H3K27me3 (Skene et al., 2018) showed that unlike that in mESCs, *Usp17l* KD resulted in mild changes of H2AK119ub1/H3K27me3 globally (Fig. R5A, left, “KD”, and R5B, the 1st and 3rd columns). *Usp17le* OE, by contrast, caused a substantial global reduction of these two histone marks (Fig. R5A, left, “OE”, and R5B, the 2nd and 4th columns), which was also confirmed by immunostaining (Fig. R5C). At the *Dux* locus, H2AK119ub1 was largely unaffected upon *Usp17l* KD (Fig. R5A, right, “KD” for “H2AK119ub1”, and R5B, the 1st column, red), but decreased markedly upon *Usp17le* OE (Fig. R5A, right, “OE” for “H2AK119ub1”, and R5B, the 2nd column, red). H3K27me3 at the *Dux* locus remained low in control, *Usp17l* KD, and *Usp17le* OE embryos compared to that in mESCs (Fig. R5A-R5B, right), suggesting that H3K27me3 may not be a major repressor for *Dux* in mouse embryos. The decrease of H2AK119ub1 at the *Dux* locus upon *Usp17le* OE correlated well with the expression changes of *Dux* and other minor ZGA genes (Fig. R4G-I). Together, these results indicate that consistent with the observations in ESCs, USP17L-mediated removal of H2AK119ub1 can promote *Dux* expression and 2C program in mouse embryos

We have incorporated these new data into our revised manuscript (Fig. 6-7 in the revised manuscript,

Figure R5. The changes of H2AK119ub1 and H3K27me3 upon *Usp17i* KD and *Usp17le* OE in mouse late 2C embryos. (A) The UCSC genome browser visualization showing spike-in normalized H2AK119ub1 and H3K27me3 at *Hoxd* (left) and *Dux* (right) upon *Usp17i* KD (KD) and *Usp17le* OE (OE) in mouse late 2C embryos. H2AK119ub1 and H3K27me3 in mESCs are shown as controls. (B) Scatter plots showing the spike-in normalized read counts of H2AK119ub1 (left) and H3K27me3 (right) at gene promoters in *Usp17i* KD and *Usp17le* OE late 2C embryos. Data from two replicates are shown. *Dux* is highlighted in red. The purple dashed line indicates that the spike-in normalized read counts when they are equal between control and KD/OE L2C embryos. The black dashed line indicates that H2AK119ub1 or H3K27me3 changes at gene promoters when they are equal to that at the *Dux* locus. (C) Immunostaining showing the changes of H2AK119ub1 and H3K27me3 upon *Usp17i* KD and *Usp17le* OE in mouse late 2C embryos. Scale bar, 50 μ m.

2. Figure 3C - Various histone modifications were analyzed in *Usp17i* KD ESCs using western blot. Considering that H2BK120ub1 has been reported to stimulate H3K4me3 deposition in human cells and during ESC differentiation (Kim et al., Cell, 2009.; Chen et al., Cell Research, 2012.), it is necessary to examine the effects of *Usp17i* KD on H2BK120ub1 and H3K4me3 levels in ESCs.

Response: Thank you for this great suggestion. Western blot showed that H2BK120ub1 increased, consistent with the deubiquitination activity of USP17L, while H3K4me3 was largely unaffected in USP17L KD mESCs (Fig. R6A). In addition, H2B was detected as a potential interacting protein of USP17L, as shown in the IP-MS results (Fig. R6B), indicating that USP17L may also be involved in regulating H2BK120ub1. However, unlike H2AK119ub1, both H2BK120ub1 and H3K4me3 were low near the *Dux* locus (Fig. R6C, blue shade) (Strikoudis et al., 2017), consistent with the notion that they mark active gene bodies and promoters, respectively (Bonnet et al., 2014; Minsky et al., 2008; Rao and Dou, 2015). In addition, given that H2BK120ub1 and H3K4me3 positively correlate with transcription activity (Kim et al., 2009; Lee et al., 2007; Xu et al., 2016; Zhou et al., 2011), the increase of H2BK120ub1, if any, would be predicted to have an opposite effect (increased transcription) to the observed decrease of 2C gene expression in USP17L KD mESCs. Therefore, we do not favor the possibility that USP17L may regulate *Dux* through H2BK120ub1, although we cannot rule out the possibility that H2BK120ub1 may be involved in 2C gene regulation through other pathways.

Figure R6. H2BK120ub1 and H3K4me3 upon *Usp17l* KD in mESCs. (A) Western blot showing the levels of H2BK120ub1 and H3K4me3 in control and *Usp17l* KD mESCs. Two KD clones (KD4 and KD9) are shown. H3 is used as the loading control. (B) Scatter plot showing the USP17L-interacting proteins identified by mass spectrometry. Histone H2B is highlighted. (C) The UCSC genome browser view showing H2AK119ub1, H2BK120ub1 (Strikoudis et al., 2017), and H3K4me3 near the *Dux* locus in mESCs.

3. Figure 3C - The mechanism underlying how *Usp17l* KD results in a substantial reduction of H3K9me2/3 should be investigated. ChIP-seq of H3K9me2/3 should be analyzed in *Usp17l* KD ESCs. One possibility is that *Usp17l* KD attenuates the expression of ZGA genes, including *Zfps*, which are associated with the establishment of H3K9me2/3.

Response: Thank you for these insightful comments and suggestions. It is indeed intriguing that H3K9me2/3 decreased in USP17L KD mESCs. As suggested, we examined H3K9me3 (we did not find a suitable antibody for H3K9me2) using STAR ChIP-seq (Zhang et al., 2016) in USP17L KD mESCs. Although the global H3K9me3 patterns remained similar upon USP17L KD (Fig. R7A-B, top), spike-in normalized analyses showed that H3K9me3 was globally decreased (Fig. R7A-7B, bottom),

consistent with the western blot result (Fig. 2C in the revised manuscript). However, the expression of methyltransferases and demethylases of H3K9me2/3 was only moderately affected upon USP17L KD (Fig. R7C). There were also no significant changes in the transcription levels of ZFPs that are known to be associated with the establishment of H3K9me2/3, including *Zfp296* (Gao et al., 2023), *Zfp462* (Yelagandula et al., 2023), *Zfp51* (Xu et al., 2022), *Zfp57* (Zhang et al., 2022), *Zfp809* (Ichida et al., 2016), *Zfp961* (Yang et al., 2022) etc. (Fig. R7D). However, we cannot exclude the possibility that these factors are affected at the protein levels, which warrants further investigation.

Figure R7. The changes of H3K9me2/3 in *Usp17l* KD mESCs. (A) The UCSC genome browser visualization showing H3K9me3 in WT and *Usp17l* KD mESCs. Results before and after spike-in normalization in two KD clones are shown. (B) Scatter plots (10 kb bin) comparing H3K9me3 in WT and KD mESCs before and after spike-in normalization. (C) Bar chart showing the expression (TPM) of methyltransferases and demethylases of H3K9me2/3 in WT and *Usp17l* KD mESCs. (D) Bar chart showing the expression (TPM) of ZFPs associated with the establishment of H3K9me2/3 in WT and *Usp17l* KD mESCs.

4. Figure 4A - It would be valuable to determine whether *Usp17l* OE in *Dux* KD ESCs leads to increased expression levels of *Dux*, *Usp17l*, *Zscan4*, and other ZGA genes. Additionally, performing

mRNA-seq analysis in addition to qPCR for selective genes would enhance the comprehensiveness of the results.

Response: Thank you for these valuable comments. As suggested, we performed RNA-seq in WT, *Dux* KD, and *Dux* KD-*Usp17le* OE (*Dux* KD followed by *Usp17le* OE) mESCs (Fig. R8A). *Dux* KD resulted in substantial downregulation of 2C genes as reported (De Iaco et al., 2017; Guo et al., 2024) (Fig. R8B). The downregulation was partially rescued by *Usp17le* OE (Fig. R8B), although the final expression levels were still lower than those in WT mESCs. The expression of *Dux* was slightly increased by *Usp17le* OE (TPM from 0.05 to 0.09), presumably due to the shRNAs targeting *Dux*.

In addition, as suggested, we performed RNA-seq in WT, *Usp17le* OE, and *Usp17le* OE-*Dux* KD (*Usp17le* OE followed by *Dux* KD) mESCs besides qPCR (Fig. 3A-B in the revised manuscript). *Dux* KD partially mitigated the upregulation of 2C genes caused by *Usp17le* overexpression (Fig. R8C). These results raise a possibility that *Usp17le* may regulate 2C genes through both *Dux* dependent and independent pathways. We have incorporated these new data into our revised manuscript (Fig. 3A in the revised manuscript, page 40).

Figure R8. The expression of representative 2C genes in mESCs upon *Dux* KD and/or *Usp17le* OE. (A) Schematic showing the establishment of *Dux* KD, *Dux* KD-*Usp17le* OE, *Usp17le* OE, and *Usp17le* OE-*Dux* KD mESCs. *Dux* KD-*Usp17le* OE mESCs were established by overexpressing *Usp17le* in *Dux* KD mESCs. *Usp17le* OE-*Dux* KD mESCs were established by knocking down *Dux* in *Usp17le* OE mESCs. **(B)** Heatmap showing the changes of 2C genes in WT, *Dux* KD, and *Dux* KD-*Usp17le* OE mESCs. **(C)** Heatmap showing the changes of 2C genes in WT, *Usp17le* OE, and *Usp17le* OE-*Dux* KD mESCs. For **(B)** and **(C)**, both the raw (left in each panel) and row-normalized TPM (right in each panel) are shown.

5. Figure 5 - The authors proposed that USP17L can stabilize ZSCAN4 through direct deubiquitination. Whether USP17L can also reduce H2AK119ub1 levels to derepress Zscan4, like Dux? Examining H2AK119ub1 and H3K27me3 signals around Zscan4 loci and the expression level of Zscan4 in Usp17l KD ESCs could provide further insights into the regulatory mechanisms involved.

Response: This is a great question. We analyzed H2AK119ub1 and H3K27me3 at the *Zscan4* gene cluster. We found no significant enrichment for H2AK119ub1 and H3K27me3 (Fig. R9, “This study”), which were confirmed by a published dataset (Kundu et al., 2018) (Fig. R9, “Kundu et al.”). These results indicated that although *Zscan4* was downregulated in *Usp17l* KD mESCs at both the RNA (Fig. 1H in the revised manuscript) and protein levels (Fig. 2K in the revised manuscript), USP17L may not directly regulate *Zscan4* gene transcription through H2AK119ub1 and H3K27me3.

Figure R9. H2AK119ub1 and H3K27me3 at the *Zscan4* loci. IGV visualization of H2AK119ub1 and H3K27me3 near *Zscan4* loci in control and *Usp17l* KD mESCs. Analyses were performed using the dataset generated in this study or a published dataset (Kundu S et al., 2018).

Minor points

1. Line 202 - “Consistent with or hypothesis” may refer to “Consistent with our hypothesis”.

Response: Thank you for pointing this out. This is now corrected (line 215, page 8).

2. Figure 4C, D - The TPM values of *Dux* RNA in the two Control groups differed widely. Including a few more replicates would be beneficial to eliminate the batch effect.

Response: Thank you for this comment. We wish to note that the y-axis scales were different in the original figure, which may have caused confusion. We have now used the same scale (Fig. R10A-B), which showed largely comparable expression of *Dux* in control samples. The variation of *Dux* expression may also arise from the low expression levels of *Dux* (0.1-0.5). Despite the variation, *Dux* showed consistent expression changes between control and KD or OE mESCs (Fig. R10A-B, included as Fig. 3C-D in the revised manuscript, page 40).

Figure R10. The expression of *Dux* in mESCs upon *Usp17l* KD or *Usp17le* OE. (A) Bar chart showing the transcription level (TPM) of *Dux* after *Usp17l* knockdown in ESCs. Individual values are shown for each replicate. (B) Bar chart showing the transcription levels (TPM) of *Dux* after *Usp17le* overexpression in ESCs. Individual values are shown for each replicate.

Reviewer #3

The authors present a very interesting study showing that USP17 proteins functionally regulate mouse embryonic stem cell 2C-like state through deubiquitinating H2A (promoting *Dux* expression) and ZSCAN4 (preventing its proteasomal degradation). The manuscript is well written and the experiments are logically designed, although more information in the Methods section is required (as outlined below).

Response: Thank you for these positive and encouraging comments and suggestions. We have revised the manuscript as suggested.

1. The title of the paper is a little lacking – “USP17L promotes 2-cell-like state through deubiquitination of H2A and ZSCAN4” or something similar might be more informative.

Response: Thank you very much for this constructive comment. We have changed the title of the article to “USP17L promotes the 2-cell program through deubiquitination of H2AK119ub1 and ZSCAN4” as suggested.

2. Given the genetic complexity of the USP17-like family, it may be helpful to also refer to the other/previous names assigned to murine family members when they are first introduced in the Results section e.g. *Usp17la* (Dub1), *Usp17lb* (Dub1a) *Usp17lc* (Dub2), *Usp17ld* (Dub2a), *Usp17le* (Dub3).

Response: Thank you very much for the suggestion. We have now referred to the other/previous names of *Usp17l* genes when they were first introduced in the revised manuscript (lines 101-102, page 4).

3.1 There is some key relevant USP17 literature that at the very least needs citing, and could be discussed in the context of the authors’ data. Esrrb has been shown to upregulate USP17

expression in mESCs (which is also influenced by coactivator NCoA1 - 10.1016/j.molcel.2013.10.003, journal. pone.0093663). This is interesting as *Esrrb* is found to be upregulated following USP17 depletion in this manuscript.

Response: Thank you for your valuable suggestions. We have now cited these papers and discussed them in the revised manuscript (lines 161-166, page 6). Details are as follows:

Of note, our data show that *Esrrb* was upregulated upon *Usp17l1* depletion and downregulated upon *Usp17le* overexpression in mESCs (Fig. R11, and Fig. 1H in the manuscript). As *ESRRB* can upregulate *Usp17le* in coordination with the coactivator *NcoA1* in mESCs (van der Laan et al., 2014; van der Laan et al., 2013), these data suggest possible negative regulatory feedback between *Esrrb* and *Usp17le* in mESCs.

Figure R11. The expression of *Esrrb* in mESCs upon *Usp17l1* KD or *Usp17le* OE. Bar plots showing the expression of *Esrrb* in *Usp17l1* KD (left) and *Usp17le* OE (right) mESCs. Two clones (KD4 and KD9) for *Usp17l1* KD mESCs and the expression (TPM) from two replicates for each experiment are shown.

3.2 Furthermore, histone deubiquitination by USP17 has also previously been reported in human cells (H2AX), which was found to be important in maintaining genome integrity through regulation of DDR (10.1016/j.molonc.2014.03.003).

Response: Thank you for pointing this out. As suggested, we have included the citation in the revised manuscript, as follows:

The DUB/USP17 family proteins were initially identified in mice as deubiquitination enzymes that are involved in cell growth and viability, DNA damage response, and embryogenesis (Baek et al., 2012; Burrows et al., 2005; Delgado-Diaz et al., 2014) (lines 85-88, page 4).

4. There seem to be antibodies missing from the Key Resources Table – most notably the USP17 antibody.

Response: We apologize for the missing information. We have included related antibody information in the revised manuscript (lines 560-562, page 21), as follows: Anti-USP17L polyclonal antibody against a short peptide HRQSEPTSEDSSPIC shared by USP17LA-E was produced by GenScript.

5. There appears to be no information in the Methods section on which overexpression or knockdown plasmids were used in the study.

Response: Thank you very much for pointing this out. We have included the related information to the revised manuscript and highlighted them in blue (line 482, page 19, and line 502, page 20).

6. In the Methods section “Mutation of USP17LE”, while the mutants are defined, there should be information on how the mutagenesis was carried out here – was a commercial kit used?

Response: We appreciate this suggestion. We have included the information of how the mutagenesis was performed in the revised manuscript (lines 518-522, page 20). The sequences encoding the truncated USP17LE were amplified using specific primers (Table S4) by PCR. In addition, the mutated bases were introduced into the primers, followed by overlap extension PCR to amplify the DNA encoding USP17LE containing the mutated sites (USP17LE^{D453/457}).

7. There are two cell lines used in the study, but there appears to be very little information on which cell line was used for which experiments in either the Methods section or the Figure legends.

Response: We apologize for not clarifying this in the manuscript. The E3 ES cell line was maintained on MEF feeder cells and was initially used in this study for *Usp17l* KD and OE experiments (Fig. 1-2, 4B-D, S2). The J1 ES cell line, which can be maintained under feeder-free conditions, was subsequently used for easy manipulation. We confirmed that regulation of 2C gene expression by *Usp17l* was reproducible in J1 ES cells (Fig. 4A). Importantly, J1 cells contain stable transgene integrations of *Zscan4* promoter-driven tdTomato (Fig. 1C), which allowed for 2CLC cell sorting for subsequent experiments, including immunofluorescence (Fig. 1L), mutation experiments (Fig. 4 and S5), flow analysis (Fig. 1C-D, S1A-B, S5C), and ubiquitination experiments (Fig. 2E and 4E). We have included the cell line information in EXPERIMENTAL MODEL AND SUBJECT DETAILS in the revised manuscript (lines 464-467, pages 18-19), as follows: The E3 ES cell line was used for *Usp17l* KD and OE experiments. The J1 ES cell line, which carries the *Zscan4*-tdTomato reporter, was used for mutation experiments, flow analysis, and ubiquitination experiments.

8. In Figure 3B it is unclear where these sequences have been derived from – is USP17L supposed to represent USP17LE? Some protein identifiers in the legend would be helpful.

Response: We apologize for not clarifying this. USP17L stands for USP17LE. We have included the related information in the revised figure legend and replaced “USP17L” with “USP17LE” in Fig. 2B (former Fig. 3B) in the revised manuscript (line 1071, pages 37-38).

9. In Figures 4C, 5F and 5K, r1 and r2 should be defined – presumably replicate 1 and 2?

Response: Thank you for pointing this out. r1 and r2 indeed denote replicate 1 and 2. We have now changed “r1” and “r2” into “Rep1” and “Rep2” (page 42).

10. In Figure 5E, it is worth noting that USP17LE overexpression appears to have a global effect on ubiquitination (based on Input lanes). This has previously been reported with overexpression of human USP17 (10.1038/cr.2010.41), showing that murine family members are also very active enzymes and there are likely to be global cellular impacts from ectopic expression of USP17LE.

Response: Thank you for this valuable comment. Indeed, while we focused on the function of USP17L in regulating H2AK119ub1 and ZSCAN4 deubiquitination, the global ubiquitination level appeared to decrease upon USP17LE OE as you pointed out (Fig. 4E, the 2nd lane in the revised manuscript). We cannot rule out the possibility that USP17LE regulates other histone or non-histone targets, which may also contribute to minor ZGA gene changes. We have now discussed this in the revised manuscript (lines 301-303, page 10).

References

- Blackledge, N.P., and Klose, R.J. (2021). The molecular principles of gene regulation by Polycomb repressive complexes. *Nat Rev Mol Cell Biol* 22, 815-833.
- Bonnet, J., Wang, C.Y., Baptista, T., Vincent, S.D., Hsiao, W.C., Stierle, M., Kao, C.F., Tora, L., and Devys, D. (2014). The SAGA coactivator complex acts on the whole transcribed genome and is required for RNA polymerase II transcription. *Genes Dev* 28, 1999-2012.
- Chen, Z., and Zhang, Y. (2019). Loss of DUX causes minor defects in zygotic genome activation and is compatible with mouse development. *Nat Genet* 51, 947-951.
- De Iaco, A., Planet, E., Coluccio, A., Verp, S., Duc, J., and Trono, D. (2017). DUX-family transcription factors regulate zygotic genome activation in placental mammals. *Nat Genet* 49, 941-945.
- De Iaco, A., Verp, S., Offner, S., Grun, D., and Trono, D. (2020). DUX is a non-essential synchronizer of zygotic genome activation. *Development* 147.
- Gao, L., Zhang, Z., Zheng, X., Wang, F., Deng, Y., Zhang, Q., Wang, G., Zhang, Y., and Liu, X. (2023). The Novel Role of Zfp296 in Mammalian Embryonic Genome Activation as an H3K9me3 Modulator. *Int J Mol Sci* 24.
- Genet, M., and Torres-Padilla, M.E. (2020). The molecular and cellular features of 2-cell-like cells: a reference guide. *Development* 147.
- Grow, E.J., Weaver, B.D., Smith, C.M., Guo, J., Stein, P., Shadle, S.C., Hendrickson, P.G., Johnson, N.E., Butterfield, R.J., Menafra, R., *et al.* (2021). p53 convergently activates Dux/DUX4 in embryonic stem cells and in facioscapulohumeral muscular dystrophy cell models. *Nat Genet* 53, 1207-1220.
- Guo, M., Zhang, Y., Zhou, J., Bi, Y., Xu, J., Xu, C., Kou, X., Zhao, Y., Li, Y., Tu, Z., *et al.* (2019). Precise temporal regulation of Dux is important for embryo development. *Cell Res* 29, 956-959.
- Guo, Y., Kitano, T., Inoue, K., Murano, K., Hirose, M., Li, T.D., Sakashita, A., Ishizu, H., Ogonuki, N., Matoba, S., *et al.* (2024). Obox4 promotes zygotic genome activation upon loss of Dux. *Elife* 13.
- Hendrickson, P.G., Dorais, J.A., Grow, E.J., Whiddon, J.L., Lim, J.W., Wike, C.L., Weaver, B.D., Pflueger, C., Emery, B.R., Wilcox, A.L., *et al.* (2017). Conserved roles of mouse DUX and human DUX4 in activating cleavage-stage genes and MERVL/HERVL retrotransposons. *Nat Genet* 49, 925-934.
- Huang, Z., Yu, J., Cui, W., Johnson, B.K., Kim, K., and Pfeifer, G.P. (2021). The chromosomal protein SMCHD1 regulates DNA methylation and the 2c-like state of embryonic stem cells by antagonizing TET proteins. *Sci Adv* 7.
- Ichida, Y., Utsunomiya, Y., and Onodera, M. (2016). Effect of the linkers between the zinc fingers in zinc finger protein 809 on gene silencing and nuclear localization. *Biochem Biophys Res Commun* 471, 533-538.
- Ji, S., Chen, F., Stein, P., Wang, J., Zhou, Z., Wang, L., Zhao, Q., Lin, Z., Liu, B., Xu, K., *et al.* (2023). OBOX regulates mouse zygotic genome activation and early development. *Nature* 620, 1047-1053.
- Kim, J., Guermah, M., McGinty, R.K., Lee, J.S., Tang, Z., Milne, T.A., Shilatifard, A., Muir, T.W., and Roeder, R.G. (2009). RAD6-Mediated transcription-coupled H2B ubiquitylation directly stimulates H3K4 methylation in human cells. *Cell* 137, 459-471.
- Kundu, S., Ji, F., Sunwoo, H., Jain, G., Lee, J.T., Sadreyev, R.I., Dekker, J., and Kingston, R.E. (2018). Polycomb Repressive Complex 1 Generates Discrete Compacted Domains that Change during Differentiation. *Mol Cell* 71, 191.
- Lee, J.S., Shukla, A., Schneider, J., Swanson, S.K., Washburn, M.P., Florens, L., Bhaumik, S.R., and Shilatifard, A. (2007). Histone crosstalk between H2B monoubiquitination and H3 methylation mediated by COMPASS. *Cell* 131, 1084-1096.
- Liu, B., Xu, Q., Wang, Q., Feng, S., Lai, F., Wang, P., Zheng, F., Xiang, Y., Wu, J., Nie, J., *et al.* (2020). The landscape of RNA Pol II binding reveals a stepwise transition during ZGA. *Nature* 587, 139-144.
- Lu, F., and Zhang, Y. (2015). Cell totipotency: molecular features, induction, and maintenance. *Natl Sci Rev* 2, 217-225.
- Mei, H., Kozuka, C., Hayashi, R., Kumon, M., Koseki, H., and Inoue, A. (2021). H2AK119ub1 guides maternal inheritance

and zygotic deposition of H3K27me3 in mouse embryos. *Nat Genet* 53, 539-550.

Minsky, N., Shema, E., Field, Y., Schuster, M., Segal, E., and Oren, M. (2008). Monoubiquitinated H2B is associated with the transcribed region of highly expressed genes in human cells. *Nat Cell Biol* 10, 483-488.

Rao, R.C., and Dou, Y. (2015). Hijacked in cancer: the KMT2 (MLL) family of methyltransferases. *Nat Rev Cancer* 15, 334-346.

Rong, Y., Zhu, Y.Z., Yu, J.L., Wu, Y.W., Ji, S.Y., Zhou, Y., Jiang, Y., Jin, J., Fan, H.Y., Shen, L., *et al.* (2022). USP16-mediated histone H2A lysine-119 deubiquitination during oocyte maturation is a prerequisite for zygotic genome activation. *Nucleic Acids Res* 50, 5599-5616.

Skene, P.J., Henikoff, J.G., and Henikoff, S. (2018). Targeted in situ genome-wide profiling with high efficiency for low cell numbers. *Nat Protoc* 13, 1006-1019.

Strikoudis, A., Lazaris, C., Ntziachristos, P., Tsirigios, A., and Aifantis, I. (2017). Opposing functions of H2BK120 ubiquitylation and H3K79 methylation in the regulation of pluripotency by the Paf1 complex. *Cell Cycle* 16, 2315-2322.

van der Laan, S., Golfetto, E., Vanacker, J.M., and Maiorano, D. (2014). Cell cycle-dependent expression of Dub3, Nanog and the p160 family of nuclear receptor coactivators (NCoAs) in mouse embryonic stem cells. *PLoS One* 9, e93663.

van der Laan, S., Tsanov, N., Crozet, C., and Maiorano, D. (2013). High Dub3 expression in mouse ESCs couples the G1/S checkpoint to pluripotency. *Mol Cell* 52, 366-379.

Xiong, Z., Xu, K., Lin, Z., Kong, F., Wang, Q., Quan, Y., Sha, Q.Q., Li, F., Zou, Z., Liu, L., *et al.* (2022). Ultrasensitive Ribo-seq reveals translational landscapes during mammalian oocyte-to-embryo transition and pre-implantation development. *Nat Cell Biol* 24, 968-980.

Xu, R., Li, S., Wu, Q., Li, C., Jiang, M., Guo, L., Chen, M., Yang, L., Dong, X., Wang, H., *et al.* (2022). Stage-specific H3K9me3 occupancy ensures retrotransposon silencing in human pre-implantation embryos. *Cell Stem Cell* 29, 1051-1066 e1058.

Xu, Z., Song, Z., Li, G., Tu, H., Liu, W., Liu, Y., Wang, P., Wang, Y., Cui, X., Liu, C., *et al.* (2016). H2B ubiquitination regulates meiotic recombination by promoting chromatin relaxation. *Nucleic Acids Res* 44, 9681-9697.

Yang, B., Fang, L., Gao, Q., Xu, C., Xu, J., Chen, Z.X., Wang, Y., and Yang, P. (2022). Species-specific KRAB-ZFPs function as repressors of retroviruses by targeting PBS regions. *Proc Natl Acad Sci U S A* 119, e2119415119.

Yang, J., Cook, L., and Chen, Z. (2024). Systematic evaluation of retroviral LTRs as cis-regulatory elements in mouse embryos. *Cell Rep* 43, 113775.

Yelagandula, R., Stecher, K., Novatchkova, M., Michetti, L., Michlits, G., Wang, J., Hofbauer, P., Vainorius, G., Pribitzer, C., Isbel, L., *et al.* (2023). ZFP462 safeguards neural lineage specification by targeting G9A/GLP-mediated heterochromatin to silence enhancers. *Nat Cell Biol* 25, 42-55.

Zhang, B., Zheng, H., Huang, B., Li, W., Xiang, Y., Peng, X., Ming, J., Wu, X., Zhang, Y., Xu, Q., *et al.* (2016). Allelic reprogramming of the histone modification H3K4me3 in early mammalian development. *Nature* 537, 553-557.

Zhang, H., Li, Y., Ma, Y., Lai, C., Yu, Q., Shi, G., and Li, J. (2022). Epigenetic integrity of paternal imprints enhances the developmental potential of androgenetic haploid embryonic stem cells. *Protein Cell* 13, 102-119.

Zhang, W., Chen, F., Chen, R., Xie, D., Yang, J., Zhao, X., Guo, R., Zhang, Y., Shen, Y., Goke, J., *et al.* (2019). Zscan4c activates endogenous retrovirus MERVL and cleavage embryo genes. *Nucleic Acids Res* 47, 8485-8501.

Zheng, H., Huang, B., Zhang, B., Xiang, Y., Du, Z., Xu, Q., Li, Y., Wang, Q., Ma, J., Peng, X., *et al.* (2016). Resetting Epigenetic Memory by Reprogramming of Histone Modifications in Mammals. *Mol Cell* 63, 1066-1079.

Zhou, V.W., Goren, A., and Bernstein, B.E. (2011). Charting histone modifications and the functional organization of mammalian genomes. *Nat Rev Genet* 12, 7-18.

AUTHORS' RESPONSES TO REVIEW

We deeply appreciate the reviewers finding their comments well addressed in the revised manuscript and for additional valuable comments and suggestions. Here, we include a letter to address the remaining issues raised in the previous submission. Please note that we used Fig. 1, 2, 3, etc. to refer to figures in the revised manuscript and Fig. R1, R2, R3, etc. to refer to figures in this letter.

For the reviewers' convenience, we also included a version of the manuscript in which the revised sections related to our responses to the reviewers' comments are marked with blue.

POINT-BY-POINT RESPONSES

Reviewer #1:

1. The revised manuscript has satisfactorily addressed all my comments. This is a very nice work that, for the first time, demonstrates the link between H2Aub1 and 2C program. Although it seemed that the KD experiment in early embryos unfortunately gave negative data, I am sure that these data will also be highly appreciated in the field.

A few minor comments to be corrected before publication:

Response: We greatly appreciate these positive and encouraging comments.

2. Fig 6E. Please quantify H2Aub1 and H3K27me3 signals.

Response: We thank the reviewer for this suggestion. The signals have been quantified and included in the revised manuscript (Fig. R1A-B, Fig. 6E-F in the revised manuscript).

Figure R1. The changes of H2AK119ub1 and H3K27me3 upon *Usp17l* KD and *Usp17le* OE in mouse late 2C embryos. IF staining (A) and Jitter plot (B) showing the changes of H2AK119ub1 and H3K27me3 upon *Usp17l* KD and *Usp17le* OE in mouse late 2C embryos. The signal intensity of the control was set to 1.0. Bar=mean \pm SD. *** $P < 0.001$, ** $P < 0.01$ (two-tailed Student's t-test).

3. Fig.5, 6E. The scales of ChIP-seq signals are wanted, if possible.

Response: As suggested, we have added the scales to Fig. 5 and 6C in the revised manuscript (Fig. R2).

Figure R2. H2AK119ub1 and H3K27me3 at the *Dux* locus in WT embryos and late 2-cell embryos upon *Usp17i* KD (KD) and *Usp17le* OE (OE).

4. Fig.6, 7. In *Usp17i* KD or OE experiment, please mention what their respective controls are? (Are they non-injected embryos?)

Response: We thank the reviewer for this suggestion. We used embryos injected with water as the control for both KD and OE experiments. We have now mentioned this in the revised Methods (lines 609-610, page 20 in the revised manuscript).

5. Line 239-285 should be shortened.

Response: The original text has been shortened (lines 200-234 in the revised manuscript).

6. Line 361-379 should be moved to discussion.

Response: As suggested, the original text has been moved to the discussion part (lines 398-411, page 14 in the revised manuscript).

Reviewer #3:

The authors have addressed all of my comments in the revised manuscript.

Response: We sincerely appreciate the reviewer's previous comments, which helped substantially improve the paper.

Reviewer #4:

1. Based on the review of the manuscript and the rebuttal letter, particularly the response to Reviewer 2, I believe that the authors still have considerable work to do in addressing the reviewers' concerns. The authors have performed additional analyses using publicly available datasets and experimental validations to clarify the role of USP17L in regulating DUX and epigenetic modifications in mouse embryos, as well as other related analyses.

However, the new study, instead of supporting a definitive role for USP17L in mouse early embryos, underscores the contrasting effects of USP17 knockdown in embryonic stem cells (ESCs) versus embryos. Specifically, USP17L appears to regulate 2C-like gene expression and developmental programs differently in these two contexts. In ESCs, USP17L knockdown led to a downregulation of 2C genes, such as *Zscan4*, *Dux*, and retrotransposons, while upregulating pluripotency genes like *Nanog* and *Oct4*. In contrast, in embryos, USP17L knockdown had a much milder effect on H2AK119ub1 and did not significantly impact pre-implantation development or major ZGA, which contradicts the claims made in the manuscript. This discrepancy is particularly relevant to the manuscript's title and its major claims. Additionally, overexpression of USP17L in embryos may artificially reduce H2AK119ub1 and activate 2C genes, further raising concerns about the physiological relevance of these findings. These divergent effects question the extent to which conclusions drawn from ESC models are applicable to actual embryonic development.

Response: We thank the reviewer for these insightful comments. It is true that USP17L KD in mouse embryos did not fully reproduce the effect observed in ESCs. While USP17L may function differently in these two biological contexts, it is also possible that the impact of USP17L KD in mouse embryos is compensated by maternal USP16. USP16, which is known to deubiquitinate H2AK119ub1 (Ai et al., 2024), is highly expressed in oocytes/early embryos but is barely expressed in ESCs (Rong et al., 2022) (Fig. R3), as we discussed in the manuscript (lines 400-403, page 14 in the revised manuscript). Indeed, genetic redundancy is a major strategy for key regulatory genes in mouse early embryos. For example, *Obox* (Royall et al., 2018; Wilming et al., 2015), *Dux* (De Iaco et al., 2017; Hendrickson et al., 2017; Whiddon et al., 2017; Zhong and Holland, 2011), *Zscan4* (Falco et al., 2007), and *Usp17l* itself (Burrows et al., 2010) expressed in minor ZGA are all multi-copy genes. In the case of *Obox* genes, over 60 copies are expressed maternally or zygotically in mice to redundantly safeguard ZGA and pre-implantation development (Ji et al., 2023). DUX4 was initially thought to be non-essential for mouse development as DUX4 knockout mouse is viable (Bosnakovski et al., 2021; Chen and Zhang, 2019; De Iaco et al., 2020; Guo et al., 2019). However, it is now realized that DUX4 may be redundant with OBOX to regulate mouse ZGA and pre-implantation development (Guo et al., 2024).

While the exact reasons for the different impacts in ESCs and embryos by USP17L warrant further investigation, we felt that both findings are worth reporting, an opinion echoed by Reviewer #1, who commented that "these (embryo) data will also be highly appreciated in the field". We have revised the abstract and manuscript to refine our claims, emphasized that the conclusions are primarily based on ESC

results, and discussed the potential context-dependent functions of USP17L. The details are as follows: “Here, through knockdown, overexpression, and biochemical analyses, we find that one family of such 2C genes, *Usp17l*, plays essential roles in both transcriptional and post-translational regulation of the 2C-like state in mESCs” (Abstract, lines 36-39, page 2 in the revised manuscript); “However, we cannot exclude the possibility that USP17L functions differently in these two contexts. The conclusion that USP17L regulates the 2C program both at the transcriptional and post-translational levels is primarily based on mESC results. In sum, our findings identify USP17L as a key regulator of the 2C genes in mESCs, paving the way for future studies of the regulatory circuitry underlying the 2C program, ZGA, and totipotency” (lines 411-417, page 14 in the revised manuscript). We have also replaced “minor ZGA genes” with “2C genes” in the paper, which is commonly used to refer to genes activated in mESCs and 2CLCs (Macfarlan et al., 2012). In the title, we also emphasized “2-cell-like program” to reflect this point.

Figure R3. The expression of *Usp16* in mouse oocytes, pre-implantation embryos, and ESCs. Line charts showing the expression of *Usp16* in mouse oocytes, pre-implantation embryos, and ESCs. RNA-seq and Ribo-seq (indicating translation) data are from published data (Xiong et al., 2022; Zhang et al., 2016).

2. Beyond this major concern, further clarity is needed on the details of the immunoprecipitation experiments, especially those involving histone modifications. It is important to note that histones do not exist individually in solutions but as part of nucleosomes, which are not typically released under routine lysis conditions. Therefore, the methods used to extract and study histone modifications need to be described in detail to ensure the validity and reproducibility of the findings.

Response: We thank the reviewer for these great comments and suggestions. For the interactions between USP17L and histone modifications, Reviewer 5# also raised a similar concern. Please refer to our response to Comment 2 raised by Reviewer 5# on page 5 in this letter.

Reviewer #5:

1. The manuscript “USP17L promotes the 2-cell program through deubiquitination of H2AK119ub1 and ZSCAN4” describes the functional role of *Usp17l* during minor ZGA by deubiquitinating H2AK119ub1 and promoting the expression of *Dux*, and deubiquitinating and stabilizing *ZSCAN4*. I have not evaluated this manuscript in the first round, but I feel that the authors address the comments made during the initial evaluation very well. Also, I feel that the authors present a wide range of assays broadly supporting their conclusions.

Response: We are delighted to know the previous comments were well addressed and are grateful for the positive comments.

2. However, I do think that the proteomics part of the manuscript requires significant improvement to become meaningful:

- It is very worrying that the authors do not find H3 in Figure 2D in the “USP17L” condition, as this is part of the core histone to which also H2A belongs. In particular since H3 clearly shows up in Fig 2A as binding to USP17L. Also, given the strong co-localisation of H3K27me3 and H2AK119ub, it is remarkable that there are no H3K27me3 signals in the “USP17L” condition.

Response: We sincerely appreciate the reviewer's insightful observation. The reviewer is correct that histones generally exist as nucleosomes, where both H3 and H2A should be observed. In the previous submission, we used the high-salt NETN buffer with **500 mM NaCl** and 0.5% NP-40 for lysis and wash in the IP experiments to minimize non-specific contamination. It was reported that H2A-H2B dimers were partially dissociated from the histone octamers under this condition (Hoch et al., 2007; Park et al., 2004). By reducing the washing time and increasing the antibody usage, we can indeed detect weak H3K27me3 and H3 along with H2AK119ub1 in the immunoprecipitates of USP17L (Fig. R4, arrows). However, we do wish to note that the results exhibited substantial variations among different batches despite our repeated attempts (n=7), as H2AK119ub1 and H3K27me3 cannot be always pulled down by USP17L. It is possible that this is because the antibody did not pull down sufficient USP17L (Fig. R4, “USP17L”). Alternatively, the interaction between USP17L and H2AK119ub1/H3K27me3 may not be stable, as commonly observed for many enzymatic reactions. Similarly in Fig. 2A, H3 was not always enriched in USP17L immunoprecipitates. In light of the inconsistency, we felt that it is perhaps more appropriate to remove the IP-MS and Co-IP results of USP17L from the revised manuscript. We apologize for the oversight of the data robustness in the previous submission.

On the other hand, we would like to emphasize that this revision does not affect the major conclusions that USP17L regulates H2AK119ub1, which is strongly supported by USP17L KD and OE experiments using IF, Western blot, and ChIP-seq (Figs. 3, 4, and 6 in the revised manuscript). Importantly, the regulation of H2AK119ub1 by USP17L was impaired when using a catalytic mutant of USP17L (Fig. 3B). These data are also in line with the finding that USP16 can deubiquitinate H2AK119ub1 in mouse oocytes (Rong et al., 2022), together supporting a shared feature of these enzymes in regulating histones and gene expression.

Figure R4. Interaction between USP17L and H2AK119ub1/H3K27me3. Western blot showing the levels of various modified histones in the immunoprecipitates pulled down by anti-USP17L antibody in WT ESCs following a modified Co-IP protocol.

3. The LC-MS results of the USP17L co-IP are very poorly analyzed and cannot be interpreted or evaluated with the current analysis (Fig 2a). The authors should properly analyze the MS interaction results showing a volcano plot with showing log₂ fold changes over control (x-axis), and p-values (y-axis). What are the proteins that the authors find significantly interacting with USP17L?; please label these in such graph. Also, only such analysis will reveal whether indeed H2A, H3, and H4 histones are enriched as compared to background. Finally, these should show Zscan4 as interactor in this analysis, which is what the authors later claim in the manuscript.

- There is no description in the materials and methods of the MS experiments and their analysis.
- Please put an asterisk at USP17L on the SDS-PAGE of Fig 2a, so that readers can appreciate this positive control of the pulldown. Similarly, in Fig 2e, please point in the graph where to look for the relevant differences between wt and mutant overexpression (I understood in the end, but pointing to these would help).

Response: We thank the reviewer for these great suggestions. As the antibody we used for IP-MS can detect all USP17L proteins (USP17LA-E), we indicated a range of USP17L proteins in the original Fig. 2a (52-62 KD) (Fig. R5). A discernible band could be observed at the right molecular weight (Fig. R5A, red arrow), although the enrichment was weak likely due to the low expression of USP17L in WT ESCs. As described above, this figure is now removed from the revised manuscript as part of the USP17L IP-MS data. We sincerely apologize for the oversight of the data robustness.

Figure R5. Immunoprecipitation with the anti-USP17L antibody in WT mESCs. SDS-PAGE analysis and silver staining of the immunoprecipitates pulled down by an anti-USP17L antibody that can recognize all USP17L members (52-62 KD).

4. Btw. Fig 2e does not seem to be an *in vitro* assay, as outlined in the text, but an *in vivo* assay?

Response: Fig. 2e is indeed an *in vivo* assay in ESCs. We detected a reduction of H2AK119ub1 upon overexpression of WT but not the mutant USP17LE. We have now indicated the decreased H2AK119ub1 in the figure (Fig. R6, red arrow) and revised the description accordingly (line 242, page 9, and line 1087, page 37 in the revised manuscript).

Figure R6. USP17L regulates deubiquitination of H2AK119ub1. Western blot showing the levels of ubiquitin and H2AK119ub1 in the immunoprecipitates pulled down by anti-H2A antibody in control, USP17LE- and USP17LE^{C60A}-overexpressing ESCs. The heavy chain and light chain of IgG are indicated. Note the decrease of ubiquitin (blue arrow) and H2AK119ub1 (red arrow) when WT USP17LE is overexpressed.

5. The HA pulldown with flag-read-out, Fig 4e, is non-convincing, with a very faint band. The USP17LE-Zscan4 interaction should therefore be confirmed by alternative means, for which the authors can conveniently use their MS results as outlined above. If Zscan4 does not show up as interact in the MS results, further confirmations are even more essential.

Response: We thank the reviewer for these comments and suggestions. To further confirm the interaction between USP17LE and ZSCAN4, besides the existing data showing that overexpressed ZSCAN4 can pull down USP17LE (Fig. R6A, Fig. 4E in the revised manuscript), we now performed a reciprocal experiment, which showed overexpressed Flag-USP17LE could conversely pull down ZSCAN4 (Fig. R6B), further consolidating their interaction.

Figure R7. The Interaction between USP17LE and ZSCAN4. (A) Western blot showing the immunoprecipitation of USP17LE by ZSCAN4. IP was performed using anti-HA antibody followed by Western blot using anti-Ub, anti-HA, and anti-Flag antibodies. (B) Western blot showing the interaction between USP17LE and ZSCAN4. IP was performed using anti-Flag antibody in USP17LE-overexpressing ESCs. Immunoblot was performed using anti-Flag and anti-ZSCAN4 antibodies.

Once again, we are grateful for the Reviewer's critical comments that have led to the further improvement of this manuscript.

References

- Ai, H., He, Z., Deng, Z., Chu, G.C., Shi, Q., Tong, Z., Li, J.B., Pan, M., and Liu, L. (2024). Structural and mechanistic basis for nucleosomal H2AK119 deubiquitination by single-subunit deubiquitinase USP16. *Nat Struct Mol Biol* *31*, 1745-1755.
- Bosnakovski, D., Gearhart, M.D., Ho Choi, S., and Kyba, M. (2021). Dux facilitates post-implantation development, but is not essential for zygotic genome activation. *Biol Reprod* *104*, 83-93.
- Burrows, J.F., Scott, C.J., and Johnston, J.A. (2010). The DUB/USP17 deubiquitinating enzymes: a gene family within a tandemly repeated sequence, is also embedded within the copy number variable beta-defensin cluster. *BMC Genomics* *11*, 250.
- Chen, Z., and Zhang, Y. (2019). Loss of DUX causes minor defects in zygotic genome activation and is compatible with mouse development. *Nat Genet* *51*, 947-951.
- De Iaco, A., Planet, E., Coluccio, A., Verp, S., Duc, J., and Trono, D. (2017). DUX-family transcription factors regulate zygotic genome activation in placental mammals. *Nat Genet* *49*, 941-945.
- De Iaco, A., Verp, S., Offner, S., Grun, D., and Trono, D. (2020). DUX is a non-essential synchronizer of zygotic genome activation. *Development* *147*.
- Falco, G., Lee, S.L., Stanghellini, I., Bassey, U.C., Hamatani, T., and Ko, M.S. (2007). Zscan4: a novel gene expressed exclusively in late 2-cell embryos and embryonic stem cells. *Dev Biol* *307*, 539-550.
- Guo, M., Zhang, Y., Zhou, J., Bi, Y., Xu, J., Xu, C., Kou, X., Zhao, Y., Li, Y., Tu, Z., *et al.* (2019). Precise temporal regulation of Dux is important for embryo development. *Cell Res* *29*, 956-959.
- Guo, Y., Kitano, T., Inoue, K., Murano, K., Hirose, M., Li, T.D., Sakashita, A., Ishizu, H., Ogonuki, N., Matoba, S., *et al.* (2024). Obox4 promotes zygotic genome activation upon loss of Dux. *Elife* *13*.
- Hendrickson, P.G., Dorais, J.A., Grow, E.J., Whiddon, J.L., Lim, J.W., Wike, C.L., Weaver, B.D., Pflueger, C., Emery, B.R., Wilcox, A.L., *et al.* (2017). Conserved roles of mouse DUX and human DUX4 in activating cleavage-stage genes and MERVL/HERVL retrotransposons. *Nat Genet* *49*, 925-934.
- Hoch, D.A., Stratton, J.J., and Gloss, L.M. (2007). Protein-protein Förster resonance energy transfer analysis of nucleosome core particles containing H2A and H2A.Z. *J Mol Biol* *371*, 971-988.
- Ji, S., Chen, F., Stein, P., Wang, J., Zhou, Z., Wang, L., Zhao, Q., Lin, Z., Liu, B., Xu, K., *et al.* (2023). OBOX regulates mouse zygotic genome activation and early development. *Nature* *620*, 1047-1053.
- Macfarlan, T.S., Gifford, W.D., Driscoll, S., Lettieri, K., Rowe, H.M., Bonanomi, D., Firth, A., Singer, O., Trono, D., and Pfaff, S.L. (2012). Embryonic stem cell potency fluctuates with endogenous retrovirus activity. *Nature* *487*, 57-63.
- Park, Y.J., Dyer, P.N., Tremethick, D.J., and Luger, K. (2004). A new fluorescence resonance energy transfer approach demonstrates that the histone variant H2AZ stabilizes the histone octamer within the nucleosome. *J Biol Chem* *279*, 24274-24282.
- Rong, Y., Zhu, Y.Z., Yu, J.L., Wu, Y.W., Ji, S.Y., Zhou, Y., Jiang, Y., Jin, J., Fan, H.Y., Shen, L., *et al.* (2022). USP16-mediated histone H2A lysine-119 deubiquitination during oocyte maturation is a prerequisite for zygotic genome activation. *Nucleic Acids Res* *50*, 5599-5616.
- Royall, A.H., Maeso, I., Dunwell, T.L., and Holland, P.W.H. (2018). Mouse Obox and Crxos modulate preimplantation transcriptional profiles revealing similarity between paralogous mouse and human homeobox genes. *EvoDevo* *9*, 2.
- Whiddon, J.L., Langford, A.T., Wong, C.J., Zhong, J.W., and Tapscott, S.J. (2017). Conservation and innovation in the DUX4-family gene network. *Nat Genet* *49*, 935-940.
- Wilming, L.G., Boychenko, V., and Harrow, J.L. (2015). Comprehensive comparative homeobox gene annotation in human and mouse. *Database (Oxford)* *2015*.
- Xiong, Z., Xu, K., Lin, Z., Kong, F., Wang, Q., Quan, Y., Sha, Q.Q., Li, F., Zou, Z., Liu, L., *et al.* (2022). Ultrasensitive Ribo-seq reveals translational landscapes during mammalian oocyte-to-embryo transition and pre-implantation

development. *Nat Cell Biol* 24, 968-980.

Zhang, B., Zheng, H., Huang, B., Li, W., Xiang, Y., Peng, X., Ming, J., Wu, X., Zhang, Y., Xu, Q., *et al.* (2016). Allelic reprogramming of the histone modification H3K4me3 in early mammalian development. *Nature* 537, 553-557.

Zhong, Y.F., and Holland, P.W. (2011). The dynamics of vertebrate homeobox gene evolution: gain and loss of genes in mouse and human lineages. *BMC Evol Biol* 11, 169.

AUTHORS' RESPONSES TO REVIEW

We deeply appreciate the reviewers for the constructive and valuable comments and suggestions in the previous submission. Here we include a letter to address the remaining issues.

For the reviewers' convenience, we also included a version of the manuscript in which the revised sections related to our responses to the reviewers' comments are marked with blue.

POINT-BY-POINT RESPONSES

Reviewer #4:

1. The author has responded to my comments in a satisfactory and thorough manner. In my opinion, the manuscript now meets the standards for publication.

Response: We sincerely appreciate the reviewer finding the comments have been well addressed.

Reviewer #5:

1. By removing the interaction proteomics from their paper, the authors have addressed my main concern. While I feel removing this data affects the comprehensiveness of their study, I agree with the authors that it does not directly affect the main conclusions of the paper. Also my other concern has been addressed by performing the reciprocal pulldown.

Response: We sincerely thank the reviewer for the valuable comments and suggestions which greatly helped improve our manuscript in the previous revision.

2. Based on the comments of reviewer #4, I do think it would be useful if the authors additionally include a part in the discussion where they outline their view on the role of USP17L during early embryogenesis. Because USP17L can reprogram ES cells towards the 2C like state, but such event is unlikely to happen in vivo. So also taking into account the differences between the ES cells and in vivo embryos in their finding on USP17L, it would be interesting if the authors better indicate what role they actually envision for USP17L in early embryogenesis, and at what stage.

Response: We thank the reviewer for these excellent suggestions. The roles of USP17L in mouse embryos are absolutely important questions to be addressed in future studies. As suggested, we have now extended the discussion as follows:

“The roles of USP17L in mouse early embryogenesis warrant further investigation, which may require combined disruption of USP17L and other potentially redundant factors. We also expect that the targets of USP17L may not be limited to histones and ZSCAN4.” (lines 410-413, page 14)